
**Vertical wind retrieved by airborne lidar and analysis of**
**island induced gravity waves in combination with**
**numerical models and in-situ particle measurements**
**F. Chouza[1], O. Reitebuch[1], M. Jähn[2], S. Rahm[1], B. Weinzierl[1,3]**
[1]{Deutsches Zentrum für Luft- und Raumfahrt (DLR), Institut für Physik der Atmosphäre,
Oberpfaffenhofen, Germany}
[2]{Leibniz Institute for Tropospheric Research, Leipzig, Germany}
[3]{Ludwig-Maximilians-Universität München (LMU), Meteorologisches Institut, München,
Germany}
Correspondence to: F. Chouza (fernando.chouza@dlr.de)
**Abstract**
This study presents the analysis of island induced gravity waves observed by an airborne
Doppler wind lidar (DWL) during SALTRACE. First, the instrumental corrections required
for the retrieval of high spatial resolution vertical wind measurements from an airborne DWL
are presented and the measurement accuracy estimated by means of two different methods.
The estimated systematic error is below -0.05 m s$^{-1}$ for the selected case of study, while the
random error lies between 0.1 m s$^{-1}$ and 0.16 m s$^{-1}$ depending on the estimation method.
Then, the presented method is applied to two measurement flights during which the presence
of island induced gravity waves was detected. The first case corresponds to a research flight
conducted on 17 June 2013 in the Cape Verde islands region, while the second case
corresponds to a measurement flight on 26 June 2013 in the Barbados region. The presence of
trapped lee waves predicted by the calculated Scorer parameter profiles was confirmed by the
lidar and in-situ observations. The DWL measurements are used in combination with in-situ
wind and particle number density measurements, large eddy simulations (LES), and wavelet
analysis to determine the main characteristics of the observed island induced trapped waves.



**1   Introduction**
Large amounts of Saharan dust are transported every year across the Atlantic into the
Caribbean region (e.g. Prospero, 1999; Prospero et al., 2013). The Cape Verde and Barbados
islands, located along the main Saharan dust transport path, interact with the dust advective
flow through different mechanisms, including island induced gravity waves. This interaction
can give place to changes in the dust sedimentation rates, the vertical mixing, and clouds
formation among other effects (e.g. Engelmann et al., 2011; Savijärvi and Matthews, 2004;
Cui et al., 2012).
In order to provide new insights into the different processes that affect the Saharan mineral
dust during the long-range transport from the Sahara into the Caribbean, the Saharan Aerosol
Long-range   Transport   and   Aerosol-Cloud-Interaction   Experiment   (SALTRACE:
http://www.pa.op.dlr.de/saltrace) took place in June/July 2013. In the framework of
SALTRACE, 31 research flights were conducted between 10 June and 15 July 2013 by the
DLR (Deutsches Zentrum für Luft- und Raumfahrt) research aircraft Falcon, including several
flights in the Cape Verde islands region and Barbados. The payload deployed on board the
Falcon included a DWL (Doppler Wind lidar), aerosol, temperature, humidity and wind speed
in-situ sensors, and dropsondes. The measurement data set generated during the SALTRACE
campaign provides a good opportunity to study the generation of gravity waves by the Cape
Verde islands and Barbados and their interaction with the Saharan dust during its transport.
Although it is well known that gravity waves can be generated by orography (Smith, 1980;
Alexander and Grimsdell, 2013) and thermal effects (e.g. Baik, 1992; Savijärvi and
Matthews, 2004), the relative impact of these two mechanisms depend on the specific island
topography, location and atmospheric conditions. Previous large eddy simulation (LES)
studies performed in the Cape Verde region (Engelmann et al., 2011), conducted during the
SAMUM-2 campaign, showed that a flat island with the characteristics of the Santiago Island
(Cape Verde) can induce the generation of gravity waves and enhance the aerosol downward
mixing, only through its heat island effect, without taking into account its orography. Another
recent LES study (Jähn et al., 2015b) conducted in the Barbados area also revealed the
presence of island induced gravity waves on the lee side of the island and provided further
insight into the dust turbulent downward mixing, cloud generation and boundary layer
modification in the lee side of the island. In contrast with the previous case, the heating effect
and the orography were taken into account in this case.



Although several different measurement techniques were used to analyse gravity waves (e.g.
Kirkwood, et al., 2010; Kühnlein et al., 2013; Ehard et al., 2015), including, but not limited
to, ground based lidars and radars, airborne lidars and in-situ sensors, balloons and satellites,
the use of airborne Doppler wind lidars is unusual (Bluman and Hart, 1988). While horizontal
wind measurements retrieved by airborne DWLs are frequently found in the literature
(Reitebuch et al., 2001; Reitebuch et al., 2003; Weissmann et al., 2005; De Wekker et al.,
2012; Kavaya et al., 2014), only a few high resolution measurements of vertical winds are
reported (Kiemle et al., 2007; Kiemle et al., 2011; Emmit and Godwin, 2014). Usually, DWLs
provide measurements of the relative wind speed between the instrument and the sensed
atmospheric volume every second. Using a conically arranged measurement pattern and the
velocity-azimuth display (VAD) technique (Reitebuch et al., 2001), three dimensional
measurements of the wind field can be retrieved based on a few tens of measurements, which
corresponds to a spatial resolution on the order of a few kilometres for airborne platforms.
Although this is enough to resolve the main features of the horizontal wind field, a higher
spatial resolution in the vertical wind component is required to perform turbulence, eddy flux
and short-wavelength gravity wave studies.
In order to increase the spatial resolution of the vertical wind retrieval, fixed downward
(nadir) pointing measurements can be performed instead of the previously mentioned conical
scanning pattern. This allows the retrieval of one measurement every second, which is
equivalent to a spatial resolution of around 50 m to 200 m, depending on the aircraft type and
speed. Nevertheless, there are some problems associated with this technique which have to be
addressed to allow an accurate vertical wind retrieval.
The paper is organized as follows. Section 2 provides a brief description of the coherent DWL
mounted on the Falcon research aircraft of DLR during SALTRACE and an overview of the
method applied to retrieve high spatial resolution vertical wind measurements. Then, in
Section 3, the resulting data set is used in combination with in-situ observations and large
eddy simulations to analyse the generation, evolution and interaction with aerosols of island
induced gravity waves. Two different cases were analysed: one in the Cape Verde region and
the other in the Barbados Island. The measurements corresponding to both cases and
simulation results from the second case are compared in order to determine the similarities
and differences between the two cases, as well as the ability of the simulation to reproduce the





observed waves and provide a context to the in-situ and lidar measurements. Finally, Section
4 provides a summary and concluding remarks.
**2   Coherent DWL instrument**
**2.1   Instrument description**
During SALTRACE an airborne coherent DWL was deployed on board the DLR Falcon 20
research aircraft. The system, based on an instrument developed by CLR Photonics
(Henderson et al., 1993), today Lockheed Martin Coherent Technologies (LMCT), was
modified by DLR (Köpp et al., 2004) to provide airborne measurement capabilities. The
transceiver head, holding the diode pumped solid-state Tm:LuAG laser, the 10.8 cm diameter
afocal transceiver telescope, the receiver optics and detectors, and a double wedge scanner is
mounted on the front part of the passenger cabin (Fig. 1), while the laser power supply, the
cooling unit, the data acquisition and control electronics are mounted in two separated racks.
The lidar operates at a wavelength of 2.02254 μm, with a pulse full width at half maximum
(FWHM) of 400 ns, a pulse energy of 1-2 mJ, and a repetition frequency of 500 Hz. The key
system specifications are presented in Table 1.
Based on the heterodyne technique, DWLs are able to measure the projection of the relative
velocity between the lidar and wind along the laser pulse propagation direction. The
backscattered atmospheric signal frequency, which was affected by the Doppler effect, is
measured by mixing the backscatter signal with the laser source used to seed the outgoing
sensing pulse. Because the outgoing pulse is frequency shifted by 100 MHz with respect to
the seeding laser by an acousto-optical modulator (AOM), the DWL is able to resolve the
magnitude and sign of the Doppler induced frequency shift. Finally, applying the Doppler
equation, the measured frequency difference can be converted to a relative speed.
As mentioned before, the DWL has a dual wedge scanner system mounted in front of the
transceiver telescope. While single wedge scanners allow fixed line of sight (LOS) and
conical scan patterns with a fixed off-nadir angle, dual wedge scanning systems allow the
generation of arbitrary scanning patterns (Rahm et al., 2007; Käsler et al., 2010). In the case
of the DLR DWL, the dual wedge scanner system consists of two independently rotating
silicon wedges, with a wedge angle of 6° and their flat sides parallel arranged. Based on the



vector form of the Snell's law and the angular measurements $\theta_1$ and $\theta_2$ provided by the rotary
encoders attached to each wedge, the DWL pointing direction $\vec{L}_{DWL}(\theta_1, \theta_2)$ can be calculated
(Amirault et al., 1985).
Particularly in the case of airborne measurements, two operation modes are used: the conical
step and stare and the nadir pointing mode. The step and stare mode, retrieve horizontal wind
speed and direction based on the previously mentioned VAD technique (Reitebuch, 2012;
Weissmann et al., 2005; Reitebuch et al., 2001). The measurement pattern consists of a set of
up to 24 regularly spaced pointing directions $\vec{L}_{DWL}$ distributed in a conical geometry with an
accumulation time of 1 s or 2 s for each position. On the other side, for the retrieval of vertical
wind speed, the LOS vector $\vec{L}_{DWL}$ is set pointing approximately in nadir direction, and the
measurements are performed with an accumulation time of 1 s.
While horizontal wind speeds are about one order of magnitude lower than the aircraft speed
(approx. 180 m s$^{-1}$), vertical winds are usually two orders of magnitude lower. Because the
DWLs measure relative speed between the instrument and the sensed atmospheric volume,
especially accurate measurements of the aircraft speed, orientation, and DWL relative position
with respect to the aircraft have to be achieved in order to subtract the aircraft speed
component from the DWL measurement. A second problem associated with this measurement
scheme is the projection of the horizontal wind speed. A deviation of the DWL LOS from the
nadir direction introduces a projection of the horizontal wind speed in the vertical wind
measurement that can be only partially corrected.
Although the vertical wind retrieval would require a constant nadir pointing LOS in order to
avoid the projection of the horizontal wind speed, the system configuration during
SALTRACE did not perform an automatic correction of the LOS. However, a manual
adjustment of the LOS was performed during flight to partially correct the pitch angle of the
aircraft. The $\vec{L}_{DWL}$ vector was set with an offset angle between -2° and -2.5° around the Y axis
to partially compensate, in combination with the DWL mounting angle, the aircraft pitch
(normally between 4° and 6°).
**2.2   Calculation of the LOS pointing direction**
Based on the angles read by the scanner encoders, the output beam direction with respect to
the DWL frame can be determined. Nevertheless, because the system is mounted on an



aircraft which changes its orientation during the flight, additional transformations are required
to relate the lidar LOS vector to an Earth-fixed reference frame.
As can be seen in Fig. 1, the DWL transceiver head is mounted on the front part of the DLR
Falcon 20 research aircraft, with the transceiver telescope pointing downwards to allow the
measurement of vertical profiles. The orientation of the transceiver head with respect to the
aircraft frame (IRS frame) can be described by a set of Euler angles
$\vec{\theta}_{DWL} = [r_{DWL}, p_{DWL}, y_{DWL}]$ which are determined by the mechanical mounting of the lidar,
where $r$ is the roll angle, $p$ the pitch angle and $y$ the yaw angle. Although the magnitude of
these angles is small (on the order of a few degrees), they have to be taken into account to
avoid large systematic errors in the wind retrieval algorithm. A second set of Euler angles
$\vec{\theta}_{IRS} = [r_{IRS}, p_{IRS}, y_{IRS}]$, measured by the IRS of the aircraft, describe the orientation of the
aircraft with respect to a local NED (North-East-Down) Earth reference frame.
The LOS vector $\vec{L}_{DWL}$ calculated based on the angles measured by the scanner encoders can
be translated to a reference frame fixed to the Earth applying the following equation

$$\vec{L} = \begin{bmatrix} L_N \\ L_E \\ L_D \end{bmatrix} = C_{NED}^{IRS}(\vec{\theta}_{IRS}) \cdot C_{IRS}^{DWL}(\vec{\theta}_{DWL}) \cdot \vec{L}_{DWL}(\theta_1, \theta_2) \qquad (1)$$

where $\vec{L}$ is the LOS vector $\vec{L}_{DWL}$ referred to the Earth reference system for a given aircraft
orientation $\vec{\theta}_{IRS}$, $C_{NED}^{IRS}(\vec{\theta}_{IRS})$ is a coordinate transformation matrix between the IRS and the
NED reference frames based on the IRS measurements and the $C_{IRS}^{DWL}(\vec{\theta}_{DWL})$ is a coordinate
transformation matrix between the DWL reference frame and IRS reference frame, calculated
for a set of mounting angles $\vec{\theta}_{DWL}$ (Grewal et al., 2007).
**2.3   Vertical wind retrieval method**
For an airborne DWL with a given pointing direction, the retrieved velocity corresponds to
the relative speed, projected on the laser beam direction of propagation, between the aircraft
and the atmospheric target or ground surface contained by the sensed volume at distance R.
This relation is summarized by the equation
$v_{DWL}(R) = \vec{L} \cdot \vec{v}_{ac} + \vec{L} \cdot \vec{V}(R) + \vec{L} \cdot \vec{w}(R)$ $\qquad (2)$





Where $v_{DWL}(R)$ is the speed measured by the airborne DWL, $\vec{L}$ is the beam direction of
propagation or LOS, $\vec{v}_{ac}$ is the aircraft speed, $\vec{V}(R) = [u(R), v(R), 0]$ and $\vec{w}(R) =$
$[0, 0, w(R)]$ are the horizontal and the vertical component of the wind speed respectively. In
all cases, the vectors are referred to an Earth reference frame.
The first step in the wind retrieval process is the removal of the aircraft speed projection on
the measured LOS. While accurate measurements of the aircraft speed are obtained from a
GPS system mounted on the aircraft, the LOS vector $\vec{L}$ is determined based on the angle
measurements of the scanner encoders, the transceiver mounting orientation and the aircraft
orientation measured by the IRS.
As mentioned in the previous section, the exact transceiver head mounting angles can be
derived from DWL measurements. The proposed method for the mounting angles $\vec{\theta}_{DWL}$
estimation is based on surface returns. Previous studies used the fact that the land is immobile
to derive alignment parameters for airborne Doppler radars (Bosart et al., 2002) and lidars
(Reitebuch et al., 2001; Kavaya et al., 2014). Although this assumption is valid for the case of
land returns, in the case of sea surface returns (which is the case of most surface retrievals
during SALTRACE), the wind induced movement of the surface can introduce non
depreciable offsets in the retrievals. On the other side, deviations perpendicular to the flight
direction are hard to resolve using only surface speed measurements because this parameter is
less sensitive to rotations around the aircraft longitudinal axis (roll angle). The proposed
method for the mounting orientation estimation is based on a combination of relative surface
speeds and distances.
For the case of a range gate corresponding to land surface return, Eq. (2) is reduced to the
following expression
$v_{DWL}(R_g) = \vec{L} \cdot \vec{v}_{ac}$           (3)
As was mentioned in the previous section, the DWL mounting angles are small. Based on this
fact, a small angle approximation can be applied to the rotation matrix $C_{IRS}^{DWL}$ from Eq. (1),
resulting in the following matrix
$C_{IRS}^{DWL}(\vec{\theta}_{DWL}) \approx \begin{pmatrix} 1 & -y_{DWL} & p_{DWL} \\ y_{DWL} & 1 & -r_{DWL} \\ -p_{DWL} & r_{DWL} & 1 \end{pmatrix}$       (4)





Then, substituting Eq. (4) in Eq. (1) and the result in Eq. (3), we can write the following linear
equation
$$v_{DWL}(R_g) = C_{NED}^{IRS}(\vec{\theta}_{IRS}) \cdot \begin{pmatrix} 1 & -y_{DWL} & p_{DWL} \\ y_{DWL} & 1 & -r_{DWL} \\ -p_{DWL} & r_{DWL} & 1 \end{pmatrix} \cdot \vec{L}_{DWL}(\theta_1, \theta_2) \cdot \vec{v}_{ac} \quad (5)$$
Using a set of land surface return measurements obtained by the lidar operating in conical
scanning mode, an overdetermined set of linear equations can be defined. Its solution gives us
an estimation of the DWL mounting orientation $\vec{\theta}_{DWL}$.
Although it is possible to estimate the mounting angles based only on land surface speed
measurements, the additional use of surface distance measurements reduces the amount of
required data points and increases the accuracy of the estimation, especially with respect to
rotations of the transceiver head around the aircraft longitudinal axis $r_{DWL}$.
For a range gate corresponding to sea surface return and neglecting Earth curvature effects,
the measured distance by the lidar can be approximated by
$$d_{DWL}(R_g) = \frac{h_{ac}}{L_D} \quad (6)$$
Where $d_{DWL}(R_g)$ is the distance measured by the lidar, $h_{ac}$ is the altitude of the DWL
measured by the GPS (World Geodetic System 1984 standard, WGS84) and corrected taking
in    account    the    Earth    Gravitational    Model    1996    (EGM96,    http://earth-
info.nga.mil/GandG/wgs84/gravitymod/egm96/egm96.html),    and    $L_D$    is    the    vertical
component of the LOS vector $\vec{L}$ approximated by Eq. (5). While the use of land surface return
is also possible, it requires additional processing and the use of a DEM (Digital Elevation
Model). For the particular case of SALTRACE, for which most of the measurements were
performed over sea, only the sea surface was used as distance reference for the estimation of
the mounting orientation.
In order to characterize the stability of the mounting angles of the DWL, three different
analyses were performed. First, an analysis was done based on the observation of the changes
in the mounting angle for different flight conditions. For a given flight, the surface returns
were grouped according to the flight altitude and the mounting angles retrieved using Eq. (5)
and Eq. (6). For altitudes below 5000 m, the mounting angles showed a slight change with
respect to those retrieved for higher altitudes. This effect can be attributed to small aircraft





deformations, mounting angle variation due to the lidar flexible mounting system and the
consequent change in the orientation of the lidar with respect to the aircraft IRS.
Then, a second stability analysis between different flights was performed. For each flight, the
surface returns (land and sea) corresponding to flight altitudes higher than 5000 m, with
vertical aircraft speeds lower than 0.05 m s$^{-1}$ and performed using scanning operation mode,
were used to retrieve the mounting angles. An equation system based on Eq. (6) was defined
for the case of points retrieved from land surface, while Eq. (7) was used for the case of sea
surface returns. For flights for which more than 50 land and sea surface return measurements
where available, the equation system was solved. The results are presented in Table 2,
together with the results obtained for the same flights using only one of both return types.
As can be seen in Table 2, the estimated mounting pitch angle $p_{DWL}$ does not show a big
difference between both methods. This result is expected because both the distance and the
ground speed measurements are strongly affected by deviations in the pitch angle. On the
other hand, for the case of the roll mounting angle $r_{DWL}$, the use of only one method gives
place to different solutions between methods and flights, while the combination of both
methods gives more stable results. The yaw angle $y_{DWL}$ is better resolved by the speed
measurements, which is compatible with the expected behaviour. In general, the simultaneous
use of both methods gives more stable results between flights.
Finally, because the orientation showed to be stable between different flights, the same
method was applied using all sea and ground surface returns which fulfill the altitude, vertical
velocity and operation mode previously described, independently of the flight. The
coincidence with the flight by flight calculation using distance and velocity equations is better
than 0.1° for pitch and yaw angles, and better than 0.2° for the roll angle estimation.
The mounting orientation estimation resulting from the last calculation with all flights
$\vec{\theta}_{DWL} = [0.98°; -2.08°; 1.62°]$ was used in the horizontal and vertical wind retrieval process
for all SALTRACE flights legs flown above 5000 m. For the case of low level flight legs
(<5000 m) during which the aircraft deformation was relevant and surface returns are
available, two different approaches can be applied to correct this effect. If the surface
measurements were obtained by the lidar operating in scanning mode, it is possible to
recalculate the DWL mounting orientation based on those observations. In the other hand, if
the measurements were performed in nadir pointing mode, the mean difference between the





surface return speed measured by the lidar $v_{DWL}(R_g)$ and their estimation $\vec{L} \cdot \vec{v}_{ac}$ can be
subtracted from the retrievals corresponding to atmospheric range gates $v_{DWL}(R)$ in order to
partially compensate this effect.
For the particular case of vertical wind measurements, the LOS vector $\vec{L}$ has to be pointing
downwards during the measurements. The better this condition is fulfilled; the better will be
the retrieved vertical wind measurements. Small deviations from vertical pointing, introduces
a projection of the horizontal wind speed on the LOS which cannot be distinguished from the
vertical wind component. For example, for an horizontal wind speed of 10 m s$^{-1}$ and a
direction contrary to the flight direction, a deviation of 0.5° from nadir in the pitch axis will
introduce a bias of 0.09 m s$^{-1}$ in the measured LOS speed.
As mentioned in Sec. 2.1, because no mounting orientation characterization was performed
before the campaign and no automatic LOS correction was implemented to compensate
changes in the aircraft orientation; the vertical wind measurements include the effects of the
horizontal wind projection on the LOS $\vec{L} \cdot \vec{V}(R)$. To partially correct this effect, estimations
of the horizontal wind speed based on DWL measurements, dropsondes and models from the
same flight leg or close were used.
**2.4   Validation of method and error analysis**
In order to test and validate the method presented in the previous section, the measurements
corresponding to a flight performed on 20 June 2013 were used. As a first approach, the
mounting angles retrieved in the previous section were applied to estimate the surface return
speed for the particular case of the leg flown between 10:31 and 10:45 LT at an altitude of
2900 m during which the lidar was operating in nadir pointing mode. The resulting surface
speed measurements show a standard deviation of ~0.4 m s$^{-1}$ for sea return points and ~0.1 m
s$^{-1}$ for land return points with a systematic difference of around -0.4 m s$^{-1}$ between the
expected and the measured ground speed values. This difference, as explained before, can be
attributed to a change in the relative position of the lidar and the aircraft IRS due to aircraft
deformations during low level flights. For this case, because measurements using scanning
mode were performed during the same flight and under similar speed and altitude conditions,
new mounting angles were calculated. The major difference of the recalculated values
$\vec{\theta}_{DWL} = [0.96°; -2.24°; 1.68°]$ is in the pitch angle (0.16°), which is consistent with
measurements performed to analyze the stability of the mounting orientation during the flight.





The recalculated mean difference between the measurements and the estimations obtained
based on the new set of mounting angles is $0.036\,\mathrm{ms^{-1}}$, while the standard deviation
corresponding to land and sea returns remains the same.
From the in-situ vertical wind speed measurements of the Falcon nose boom sensors
presented in Fig. 2a, it can be seen that the Falcon was flying through gravity waves. These
waves induce a change in the aircraft pitch (Fig. 2a), which in turn induce a change in the
LOS of the lidar and, therefore, a varying horizontal wind projection with a contribution
between $\pm 0.15\,\mathrm{ms^{-1}}$ for a horizontal speed of $12\,\mathrm{ms^{-1}}$. In order to partially correct this
feedback effect, an estimation of the horizontal wind speed and direction obtained from a
previous measurement leg (10:00 to 10:15 LT) was used (Fig. 2b). The resulting projection
$\vec{L} \cdot \vec{V}$ (R) is shown in Fig. 2c.
Finally, the vertical wind speed can be determined subtracting the aircraft speed and
horizontal wind speed projections from the relative speed measured by the DWL. The
resulting vertical wind speeds are presented in Fig. 3a together with a comparison between the
in-situ vertical wind speeds measured by the Falcon (Fig.4). It has to be noted that the closest
DWL measurements are around 500 m below the aircraft, which could explain the difference
between the amplitude of the DWL and in-situ measurements. Despite this altitude difference,
the main features of the vertical wind field (amplitude, oscillation frequency and mean value)
are comparable.
In order to estimate the DWL measurement error, two methods were applied. The first method
(Frehlich, 2001; O'Connor et al., 2010) is based on the frequency spectrum of the retrieved
velocity. For the flight leg presented in Fig. 3, the spectra corresponding to the retrieved
vertical wind speed from the DWL measurements at altitude of 2300 m were calculated and
averaged. A total of 32 spectra of 64 samples were calculated. A 50% window overlap factor,
a Hanning window and a zero-padding of the missing values was applied to each window for
each spectrum calculation (Kiemle et al., 2011). The resulting spectrum, presented in Fig. 4,
shows that for frequencies higher than 0.2 Hz the spectrum corresponding to the DWL tends
to a constant value, departing from the Kolmogorov's -5/3 law. This high frequency region
represents the spectrum of the random measurement noise. The standard deviation of the
measurement is then estimated as the mean of the spectra over its constant region. Based on
the presented case, the random measurement error was estimated to be $\sigma_e = 0.16\,\mathrm{ms^{-1}}$.





The second method, based on ground return analysis, relies on the fact that the ground surface
is not moving. For an ideal system, the difference between the ground return speed measured
by the lidar and the aircraft speed projected on the beam direction has to be zero. For a real
DWL, the mean of this difference corresponds to the systematic measurement error, while its
variations correspond to the lidar random error for high SNR. Based on the land returns, the
estimated systematic error is $\mu_e = -0.05 \text{ m s}^{-1}$ and random error standard deviation is
$\sigma_e = 0.10 \text{ ms}^{-1}$.
The errors estimated based on both approaches are on the same order of magnitude. Because
the measurement error depends on many different parameters, like SNR, turbulence and flight
conditions, relatively small variations in the uncertainty are expected for differing
measurement situations.
**3   Case studies**
The vertical wind retrieval method presented in the previous section was applied to the DWL
measurements corresponding to two SALTRACE flights during which gravity waves were
observed. While the flight corresponding to the first of case study took place on 17 June 2013
in the Cape Verde region (close to the Saharan dust source), the flight analysed in the second
case study was conducted in the Barbados region, where the main SALTRACE supersite was
located.
**3.1   Case 1: Island induced trapped lee waves in Cape Verde (17 June 2013)**
The synoptic conditions, derived from the ECMWF Era-Interim reanalysis, on the Cape
Verde islands at 12:00 UTC (11:00 LT) on 17 June 2013 are show for 1000 hPa and 700 hPa
pressure levels in Fig. 5 (upper panels). Northerly trade winds with a speed on the order of 5-
10 m s$^{-1}$ are visible at the lower pressure level. For the upper pressure level the wind changes
to easterly direction, which is compatible with the presence of the African Easterly Jet (AEJ).
During the morning of 17 June 2013, the Falcon performed a measurement flight in the Cape
Verde islands region, departing from Sal and landing on Praia (Fig. S1). The flight consisted
of three legs between Sal and Praia islands, at altitudes of 4100 m, 2500 m and 900 m
respectively. During the ascent phase of the flight, the in-situ sensors on board the aircraft
recorded vertical profiles of potential temperature, wind speed and direction (Fig 6a). The
horizontal winds retrieved by the DWL during segments of the 1$^{st}$ and 2$^{nd}$ leg (not shown)
indicate that the values retrieved by the in-situ sensors are representative of the whole flight





area and time period. For altitudes below 500 m, a neutrally stratified layer with northerly
trade winds can be observed. This layer is capped by a thick strongly stratified trade inversion
layer (between 500 m and 2000 m) on which the wind direction exhibits a strong shear and a
change to north-easterly direction. Above this inversion, the atmosphere shows a weak
stratification and relatively constant wind speed and direction. These observations are
compatible with the synoptic situation previously described. Based on the in-situ
measurements, a profile of the Scorer parameter was calculated (Fig 6b).
According to linear mountain wave theory (e.g. Durran, 1990), the waves that can propagate
vertically in the atmosphere can be derived by the use of the Scorer parameter $l^2$ (Scorer,
1949), given by:
$$l^2 = \frac{N^2}{U^2} - \frac{1}{U}\frac{d^2U}{dz^2} \qquad\qquad (7)$$
where N is the Brunt-Väisälä frequency and U is the cross-mountain wind speed. Since the
difficulty to estimate the second derivate of the wind from real data, an approximated form of
the Scorer parameter which neglects the second term (shear term) is used in this study. This
simplification is allowed due to the fact that the shear term is dominated by oscillations with
wavelengths much smaller than for the waves under analysis (Smith et al., 2002). An
additional validation of this approximation was performed based on temperature, humidity
and wind speed vertical profiles retrieved from the ECMWF ERA-Interim reanalysis
presented in Fig. 5. The Scorer parameter profile calculated based on the model output and
including the shear term (not shown) exhibited a similar structure to the approximated profile
calculated from the Falcon in-situ measurements.
According to the theory, a wave with wavelength $\lambda$ and an associated wave number $k = \frac{2\pi}{\lambda}$
can propagate in the atmosphere if $k^2 < l^2$; otherwise, the wave is evanescent. Trapped lee
waves are expected when a layer with high Scorer parameter is bounded by layers with low
Scorer parameter. Under such conditions, the energy of the gravity waves is mainly confined
in the layer with high Scorer parameter (Durran et al., 2015). As can be seen in Fig. 6b, the
conditions for the development of trapped waves are fulfilled for altitudes between 500 m and
2000 m, which are coincident with the trade inversion layer lower and upper bound.
Another important parameter to take in account for the study of lee waves is the inverse
Froude number, defined as:





$F = \frac{Nh}{U}$ (8)
where N is the Brunt-Väisälä frequency, h is the island height and U is the cross-mountain
wind speed. For an inverse Froude number between 0 and 0.5, the flow behaves as described
by the linear theory and waves are expected to form in the lee side of the island (Baines and
Hoinka, 1985). For this case study, the Brunt-Väisälä frequency associated with the boundary
layer (below 500m) is approximately $N = 0.01 \, s^{-1}$. This situation leads, together with an
inflow speed of $10 \, m \, s^{-1}$ and a maximum island height of 380 m, to a linear flow condition
($F = 0.38$).
The DWL vertical wind measurements presented in Fig. 7a confirm the existence of trapped
waves between 500 m and 1500 m. At first glance, an apparent wavelength of approximately
5 km and a maximum peak-to-peak amplitude of 1.5 m s$^{-1}$ can be estimated from the
measurements presented in Fig. 7a. Unfortunately, although the legs were flown at different
altitudes, all were performed along the same track. For this reason, only the apparent
wavelength can be estimated from these measurements. This apparent wavelength is always
larger than the actual one and related to the angle defined between the flight track and the
propagation direction of the waves. Based on the apparent wavelength, the associated
apparent wave number was calculated and plotted together with the Scorer parameter profile
(Fig. 6b). It can be seen that for the layer on which the trapped waves are observed, the Scorer
condition for wave propagation $k^2 < l^2$ is fulfilled, while, for altitudes above 1500 m where
evanescent waves are expected ($k^2 > l^2$), the waves exhibit a strong reduction in their
amplitude as a function of the altitude (0.6 m s$^{-1}$ at 2100 m and 0.4 m s$^{-1}$ at 2600 m, Fig. 7c).
Together with the DWL vertical winds, a set of vertical wind speed measurements was
retrieved by the Falcon in-situ sensors along each measurement leg (Figs. 7b, c and d).
Although the vertical wind measurements from the DWL provide an better image of the
vertical extension of the wave ducting, the evanescent propagation regime, and the amplitude
of the waves than the in-situ measurements, the DWL measurements of this case study covers
only half of the 1$^{st}$ and one third of the 2$^{nd}$ leg, limiting the spectral range that can be analysed
from those measurements. In order to complement the DWL measurements and obtain a more
precise picture of the spectral components and extension of the observed waves, the wavelet
transform technique using a Morlet wavelet (6$^{th}$ order) was applied (Torrence and Compo,
1998; Woods and Smith, 2010) to the in-situ vertical wind measurements of the three
measurement legs. Unlike the Fourier transformation, which can only retrieve the frequency




of the signal, the wavelet analysis is able to temporally resolve the frequencies of a given
signal, similar to the short-time Fourier transformation. For this reason and because the
measured waves are restricted to a fraction of the measurement period the wavelet analysis is
more adequate than a simple Fourier transformation. The in-situ measurements, acquired with
a temporal resolution of 1 Hz, were linearly interpolated to a regular spatial grid of 200 m
resolution before applying the wavelet transformation to the acquired datasets.
Figure 8 shows the calculated spectral components corresponding to the three legs in-situ
vertical wind measurements. While the lower leg (Fig. 8, lower panel) exhibits one dominant
spectral component with an associated apparent wavelength of 5.2 km, the $2^{nd}$ leg (Fig. 8, mid
panel) shows two spectral peaks with apparent wavelengths of 4.5 km and 15.7 km, and the
$3^{rd}$ leg (Fig. 8, upper panel) three peaks with apparent wavelengths of 4.5 km, 7.53 km and
16.5 km. Although some variability in the wavelengths is observed between legs, the modes
seem to be in even-harmonic relation. The shorter mode, present in all three legs, shows a
strong attenuation in the upper two legs, compatible with the evanescent propagation regime.
In contrast, the longer mode, present in the upper two legs, is only slightly attenuated.
Previous studies on trapped lee waves (Georgelin and Lott, 2001) showed the presence of
upward propagating leaked harmonics above the wave duct. In contrast to that case, the
spectral analysis of the lower leg doesn't show the presence of the harmonics observed in the
upper two, which can be attributed to different reasons. On one side, the longer modes have
wavelengths comparable to the length of the measurements leg, which limits the confidence
of this observation. This limitation is represented by the cone of influence of the wavelet
transform, which indicates the region of the wavelet spectra in which edge effects become
relevant. On the other side, although the measurement legs were flown within a short time
period, changes in the atmospheric conditions could have introduced a modification in the
wave propagation and generation conditions. Finally, taking into account that the
measurement legs extend along the downwind region of the Boa Vista and Sal islands, an
interaction between waves generated by these two islands cannot be discarded.
A second point to be noted from the vertical wind in-situ measurements is the change in the
position of the wave crests and troughs between the measurements of the $2^{nd}$ and $3^{rd}$ leg,
which suggest that the waves are not completely stationary.
Together with the vertical wind speed, the DWL measured the backscattered power. Range
corrected backscattered power is shown in Fig. 9a together with in-situ vertical winds and





Condensation particle counter (CPC) measurements (Weinzierl et al., 2011) for the three legs
(Fig. 9b, c, d). The CPC measurements correspond to the number concentration of particles
with diameter larger than 14 nm. Figure 9a gives a qualitative picture of the aerosol
distribution. Below 500 m, a region of high backscatter corresponding to the marine boundary
layer can be recognized. The presence of some clouds on top of the marine boundary layer is
indicated by high backscatter regions. The interaction between the trapped waves, a thin
aerosol layer which extends between 900 m and 1300 m, and the bottom of the Saharan Air
Layer (SAL) can also be noted. The CPC measurements of the 1st and 2nd legs, corresponding
to the legs flown in the SAL, show a relatively homogeneous particle number concentration
along the leg, with slightly higher values in the upper one.
As the 3rd measurement leg was flown at an altitude (1000 m) coincident with a strong
gradient in the aerosol concentration (Fig. 9a), the CPC measurements show also the effect of
the waves in the aerosol distribution. This explains why the waves are only visible in the third
leg and not in the upper two, where the aerosol concentration gradient is close to zero. The
phase difference of 90 degrees between the aerosol concentration change and the vertical
wind velocity leads to a no net flux condition, where the dust loaded air parcels periodically
rise and sink without leading to a net downward or upward dust transport.

## 3.2  Case 2: Island induced trapped lee waves in Barbados (26 June 2013)

### 3.2.1  Measurements

During SALTRACE several flights were conducted in the Barbados region. In this study, the
measurements corresponding to the flight during the evening of the 26 June 2013 are used to
study the presence of island induced gravity waves in the lee side of Barbados (Fig. S2). In
this case, the flight path had a cross shape centred in Bridgetown city with constant longitude
(1st Leg) and latitude (2nd, 3rd and 4th Legs) at an altitude of approximately 7600 m for the first
two legs and 3500 m and 1200 m for the 3rd and 4th legs respectively.
The synoptic situation at 00:00 UTC (20:00 LT) on the 27 June 2013 is show in Fig. 5 (lower
panels) for 1000 hPa and 700 hPa pressure levels. Easterly winds with a speed of
approximately 10 m s$^{-1}$ can be seen at both pressure levels. Wind speed, direction and
potential temperature profiles derived from a dropsonde launched at the end of the second





measurement leg (Fig. S2) are presented in Fig. 10a. Below 500 m, an almost neutrally
stratified boundary layer can be recognized, which is topped by a second neutrally stratified
layer between 500 m and 1800 m. Between 1800 m and 2000 m, the sounding measurements
show a thin and strong inversion ($\Delta\theta = 6\ K$) coincident with the lower bound of the SAL,
which extends between 2000 m and 4100 m and exhibits a typical weak stratification. Along
these three layers, easterly winds with mean wind speed of 10 m s$^{-1}$ are observed, while above
the SAL, the stratification increases, the wind speed reduces and the direction reverses. Based
on these measurements, the vertical profile of the Scorer parameter in its approximated form
(Fig. 10) was calculated. As in the previous case, these measurements were compared to the
DWL horizontal wind retrieval and the ECMWF ERA-Interim reanalysis results (not shown)
to confirm the representativeness of the measurements.
As expected, the calculated Scorer profile shows a thin layer of high Scorer parameter
corresponding to the strong inversion shown at 2000 m, upper bounded by a low Scorer
parameter layer which extends up to 3800 m and is associated with the weak stratification of
the SAL. Trapped waves at the density discontinuity associated with the inversion at 2000 m
are likely to occur in such conditions (Vosper, 2004; Sachsperger et al., 2015). Above the
SAL, the Scorer parameter increases due to a decrease in the wind speed and a stronger
stability of the atmosphere. The increase in the Scorer parameter can lead to the presence of
some wave leakage into the upper layers. Because the boundary layer is very weakly
stratified, the inverse Froude number is likely to be very close to zero. Based on the
dropsonde observations, the Brunt-Väisälä frequency at the boundary layer was estimated to
be N = 0.0025 s$^{-1}$ and the cross mountain wind speed approximately 12 m s$^{-1}$, which
together with a maximum island height of 340 m gives place to $F = 0.07$, suggesting a linear
flow condition.
These expectations were confirmed by the DWL observations, which showed the presence of
trapped lee waves on the lee side of Barbados (Fig. 11). The vertical winds measured by the
DWL during the first leg (Fig. 11a) show a strong updraft with 1.2 m s$^{-1}$ on the north part of
Barbados, which is compatible with a measurement along a crest of the trapped waves. The
2$^{nd}$ leg (Fig 11b) supports this observation, showing a wave structure on the lee side, with a
wavelength of approximately 9 to 10 km and a vertical extension between 500 m and the top
of the SAL. Unfortunately the low aerosol load above the top of the SAL (4100 m) limits the
DWL coverage and the possibility to confirm the leakage by direct lidar measurements. The



maximum amplitude (up to 3 m s$^{-1}$ peak to peak) of the waves is found at an altitude of around
2000 m, coinciding with high Scorer parameter layer, the temperature inversion and the
maxima in the wind shear altitude. Above and below the inversion, the wave amplitude
decrease, which is compatible with the evanescent wave regime $k^2 > l^2$ observed in the
Scorer parameter plot (Fig. 10b).
Although in this case the DWL measurement leg is long enough to resolve the spectra of the
observed waves below the top of the SAL, the low aerosol load limits the DWL coverage in
the upper part of the troposphere. For this reason, a wavelet transform was applied to the in-
situ vertical wind measurements of the 2$^{nd}$, 3$^{rd}$ and 4$^{th}$ legs in order to determine the spectra of
the waves and whether or not there is leakage of the wave into upper layers (Fig. 12). The
spectra of the in-situ vertical wind measurements at 7745 m, corresponding to the 2$^{nd}$ leg (Fig.
11c and Fig. 12, upper panel), does not show signs of a spectral component with a wavelength
around 9 km, suggesting that this propagation mode does not leak into upper layers and that
the wave dissipation is dominated by the boundary layer absorption and dispersion (Durran et
al., 2015). This finding is compatible with the relatively short wavelength of the observed
wave, which lead to a strong vertical decay in the evanescent wave regime regions.
The spectral analysis corresponding to the 3$^{rd}$ and 4$^{th}$ legs (Fig. 12, mid and lower panel)
indicate the presence of a spectral component of the same wavelength than the observed by
the lidar during the 2$^{nd}$ leg (9.5 km), but with much lower amplitude. Although this change is
unexpected due to the short time difference between the 2$^{nd}$ leg (21:49 to 22:05 LT) and the
3$^{rd}$ leg (22:16 to 22:34 LT), relatively small changes in the atmospheric stability conditions
could have occurred in the time interval between both measurements, leading to a change in
the waves propagation condition.
Together with vertical wind measurements, calibrated backscatter coefficient profiles (Figs.
13a and 13b) were retrieved from the DWL measurements (Chouza et al., 2015) for the 1$^{st}$
and 2$^{nd}$ leg. A three layer structure can be recognized in this case, with the marine boundary
layer below 500 m, a mixed layer between 500 m and 2000 m, and the SAL between 2000 m
and 4100 m. A wave structure can be identified in the boundary between the SAL and the
mixed layer at an altitude of 2000 m. Because the 3$^{rd}$ and 4$^{th}$ legs were flown at altitudes were
the gradient in the aerosol concentration was very low, no signature of waves is observed in
this case (Figs. 13c and 13d).



The backscatter coefficient and vertical wind corresponding to the boundary were the waves
were observed is displayed in Fig. 14. As for the previous case, a phase difference of 90
degrees is observed between both quantities, which is compatible with a no net flux condition.
Although in this particular case this feature was already observed by in-situ measurements in
the previous case study, the shown measurements provide an example of the possibilities
opened up by the simultaneous retrieval of vertical wind and calibrated backscatter coefficient
from a single instrument along a whole vertical transect. Based on the technique proposed by
Engelmann et al. (2008), the simultaneous retrieval of backscatter coefficient and vertical
wind can be used, under low humidity conditions, to retrieve aerosol vertical flux profiles.

### 3.2.2 Large Eddy Simulations

Large eddy simulations are performed with the All Scale Atmospheric Model (ASAM, Jähn et
al., 2015a) for the Barbados area. The model setup is described in Jähn et al. (2015b), where
the focus of the analysis lies on island effects on boundary layer modification, cloud
generation and vertical mixing of aerosols. In this study, the simulation results along the
measurement tracks are first compared with the DWL observations in order to provide further
validation of the model and setup. Then the whole simulation results are used to provide a
broader perspective of the context within which the measurements took place.
For the simulations, a model domain with a spatial extent of 102.4 x 102.4 km is chosen with
Barbados located at the domain center. The model top is at 5 km altitude. Grid spacings of
200 m (horizontally) and 50 m (vertically) are used and topographical data is obtained from
the Consortium for Spatial Information (CGIAR-CSI) Shuttle Radar Topography Mission
(SRTM) dataset (http://srtm.csi.cgiar.org) at 200 m resolution. The highest elevation of
Barbados is Mount Hillaby, 340 m above sea level. Due to the presence of a topographically
structured island surface in the domain center, the simulations are performed with open lateral
boundaries.
In order to generate inflow turbulence consistent with the upstream marine boundary layer
forcing, the newly developed turbulence generation method is applied. The model runs are
initialized with nighttime radiosonde data of the considered day and are integrated from 02:00
to 22:00 LT. Further details on the setup and comparison with the DWL can be found in Jähn
et al. (2015b).





In order to allow a qualitative comparison of the results from the LES with the measurements
from the DWL, plots of the simulated vertical wind speed on time periods similar to those
corresponding to the measurements are presented in Figs. 15 and 16. Figure 15 shows two
horizontal cuts of the vertical wind speed at 20:00 LT and 22:00 LT together with the flight
track of the Falcon corresponding to legs 1 and 2. The horizontal plane is located at an
altitude of 1975 m, coinciding with the temperature inversion observed by the in-situ
measurements and the maximum in the horizontal wind shear observed by the DWL, in-situ
and dropsonde measurements. Figure 16 displays the simulated vertical winds corresponding
to the 1$^{st}$ and 2$^{nd}$ legs flown by the Falcon (indicated in Fig. 15). The horizontal scales of the
plots were adjusted to simplify the comparison with the measurements presented in Fig. 11.
**3.2.3  Comparison of LES and DWL**
As can be seen in Figs. 15 and 16, the LES is able to reproduce the observations from the
DWL. A limited coverage over Barbados due to the presence of convective clouds is
compatible with the convective plumes observed in the simulations. Trapped waves with a
wavelength and extension similar to the lidar observations can be recognized on the lee side
of Barbados. The vertical cut presented in Fig. 16a shows an updraft above Barbados similar
the one shown in Fig. 11a and Fig. 15a shows that the vertical cut is located along a wave
crest. This simulation result provides support to the hypothesis presented in Sec. 4.1 to
explain the strong updraft observed above Barbados during the first measurement leg.
A wavelet transform was applied to the measured and simulated vertical wind speed in order
to provide a quantitative wavelength, amplitude and extension comparison of the simulated
and measured trapped waves. The measured data was interpolated in the same way as
described in Sec. 3, matching for this case, the spatial resolution of the LES (200 m). The
results of the wavelet transforms are presented in Fig. 17.
According to the wavelet analysis, the wavelength of the measured waves is approximately 9
km, while for the case of the LES the wavelength is approximately 7.5 km. The difference can
be attributed to different reasons such as small differences in the propagation direction of the
waves. As mentioned before, the LES is initialized with a constant wind speed direction equal
to 90°. This approximation can induce differences in the direction of propagation of the
waves, which in turn induce differences in the apparent wavelength of the measurements for
the same flight track.



While the amplitude of the simulated waves is quite similar at different altitudes, the DWL measurements show a maximum in the amplitude at approximately 2000 m. This difference is evident from the results of the wavelet analysis, calculated at the altitude of this maximum in both cases. As explained in the previous section, the horizontal wind speed used for the initialization of the LES is assumed constant between 0.7 km and 3 km. Because this approximation neglects the strong shear measured at 2000 m, a difference between the simulated and the measured vertical wind speed is expected at this altitude.

## 4   Summary and conclusions

In the first section of this work, a series of instrumental corrections required for the retrieval of vertical winds from airborne DWL measurements were presented. The difference of almost two orders of magnitude between the platform speed and the measured vertical wind speed, together with the varying aircraft orientation during the flight, transforms the retrieval of vertical winds in a challenging problem. Although no control of the lidar pointing direction was active during the vertical wind measurements, the use of horizontal wind from dropsondes and in-situ measurements proved to be useful to partially compensate the effects of the horizontal wind component projection. The use of conical scanning pattern measurements and the recalculation of the lidar mounting angle based on the ground return speed and distance previous to the vertical wind measurements proved to be useful to reduce the systematic error, especially after a change in the flight altitude. The measurement uncertainties were estimated based on two different techniques. The estimated systematic error, based one measurement case, was $-0.05$ ms$^{-1}$, while the random error was between 0.1 ms$^{-1}$ and 0.16 ms$^{-1}$ depending on the technique used for the estimation.

The described methods were applied to retrieve vertical winds corresponding to two SALTRACE research flights, one in the Cape Verde region and a second one in Barbados. The measurements revealed the presence of island induced gravity waves in both cases. Vertical profiles of temperature, wind and humidity obtained from in-situ and dropsonde measurements were used to calculate a Scorer parameter profile for each measurement case. The wavelength and the vertical extension of the trapped waves observed from the DWL measurements in the Cape Verde case study were in coincidence with the retrieved Scorer parameter profile. The spatial extension, amplitude and wavelength retrieved from the in-situ vertical wind measurements provided an independent validation for the DWL observation. A



second independent observation of the particle number concentration provided an additional
confirmation.
Although for the second case study the in-situ measurement did not show the waves observed
by the DWL in a previous leg, the results of the ASAM model support the lidar observations.
The model was able to reproduce the generation of waves in the lee side of the island and
provided a context to the lidar observations, which are limited to two dimensional vertical
cuts. The amplitude and wavelength of the simulated waves were lower than the observed
ones. This can be explained by the simplifications adopted in the horizontal wind profiles
used to initialize the model, which did not reproduce the strong shears observed in the
dropsondes and in-situ measurements.

**12 Acknowledgements**

This work was funded by the Helmholtz Association under grant number VH-NG-606
(Helmholtz-Hochschul-Nachwuchsforschergruppe AerCARE). The SALTRACE campaign
was mainly funded by the Helmholtz Association, DLR, LMU and TROPOS. The
SALTRACE flights on Cape Verde were funded through the DLR-internal project VolcATS
(Volcanic ash impact on the Air Transport System). DLR Falcon aircraft in-situ data was
processed by the DLR Flight Experiment. The first author thanks the German Academic
Exchange Service (DAAD) for the financial support.





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



1    Table 1. Key parameters of the DWL

| Laser | Laser type | Solid-state Tm:LuAG |
|---|---|---|
| | Operation wavelength | 2.02254 μm |
| | Laser energy | 1-2 mJ |
| | Repetition rate | 500 Hz |
| | Pulse length (FWHM) | 400 ns |
| Transceiver | Telescope type | Off-axis |
| | Telescope diameter | 10.8 cm |
| | Transmitted polarization | Circular |
| | Detected polarization | Co-polarized |
| Scanner | Type | Double wedge |
| | Material | Fused silica |
| Data acquisition | Sampling rate | 500 MHz |
| | Resolution | 8 bits |
| | Mode | Single shot acquisition |



1   Table 2. Estimated lidar mounting angles. The first row of each flight corresponds to the

2   retrieved mounting angle using distance and speed measurements, while the second and third

3   rows correspond to the use of only distance or speed respectively.

| Date | Sea surface points (Distance method) | Land surface points (Speed method) | Estimated mounting angle [°] | | |
|---|---|---|---|---|---|
| | | | $r_{DWL}$ | $p_{DWL}$ | $y_{DWL}$ |
| | **58** | **304** | **0.91** | **-2.09** | **1.64** |
| 12.06.13 | 58 | - | 0.82 | -2.14 | 3.23 |
| | - | 304 | 0.66 | -2.10 | 1.61 |
| | **277** | **500** | **1.00** | **-2.07** | **1.63** |
| 18.06.13 | 277 | - | 1.31 | -2.13 | -1.94 |
| | - | 500 | 0.11 | -2.11 | 1.55 |
| | **60** | **136** | **0.99** | **-2.07** | **1.63** |
| 13.07.13 | 60 | - | 1.09 | -2.11 | -0.04 |
| | - | 136 | 0.89 | -2.08 | 1.63 |
| | **269** | **161** | **0.80** | **-2.13** | **1.64** |
| 14.07.13 | 269 | - | 0.20 | -2.33 | 11.52 |
| | - | 161 | 0.66 | -2.14 | 1.64 |
| | **59** | **397** | **0.92** | **-2.07** | **1.69** |
| 14.07.13 | 59 | - | 1.13 | -2.04 | 1.82 |
| | - | 397 | 0.88 | -2.07 | 1.69 |
| | **5403** | **3449** | **0.98** | **-2.08** | **1.62** |
| All flights | 5403 | - | 0.99 | -2.11 | 1.37 |
| | - | 3449 | 1.00 | -2.08 | 1.62 |





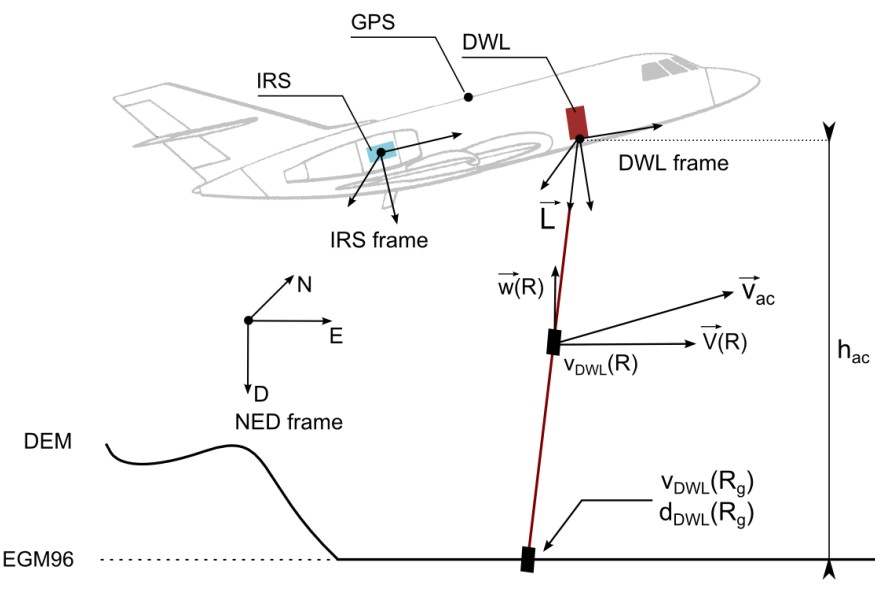

WGS84
Figure 1. Scheme with the different reference frames and magnitudes involved in the vertical
wind retrieval calculations. The DWL frame is a reference frame fixed to the DWL
transceiver head, the IRS frame is a reference frame fixed to aircraft inertial reference system
(IRS) and the NED (North-East-Down) frame is an Earth reference frame for which the x axis
is pointing northwards, the y axis is pointing eastwards and the z axis completes the right
handed reference system pointing downwards, parallel to the norm of a plane tangential to the
Earth reference ellipsoid. $\vec{L}$ is a unit vector that represents the DWL line of sight (LOS), $\vec{v}_{ac}$ is
the aircraft speed, $\vec{V}$ and $\vec{w}$ are the horizontal and vertical wind speed respectively in a range
R from the lidar, $v_{DWL}$ is the relative speed measured by the DWL, $h_{ac}$ is the aircraft altitude
about ground level and $R_g$ is the range between the lidar and the ground. WGS84 is the World
Geodetic System 1984 standard used by the GPS system, while EGM96 is the Earth
Gravitational Model 1996 used for correction.



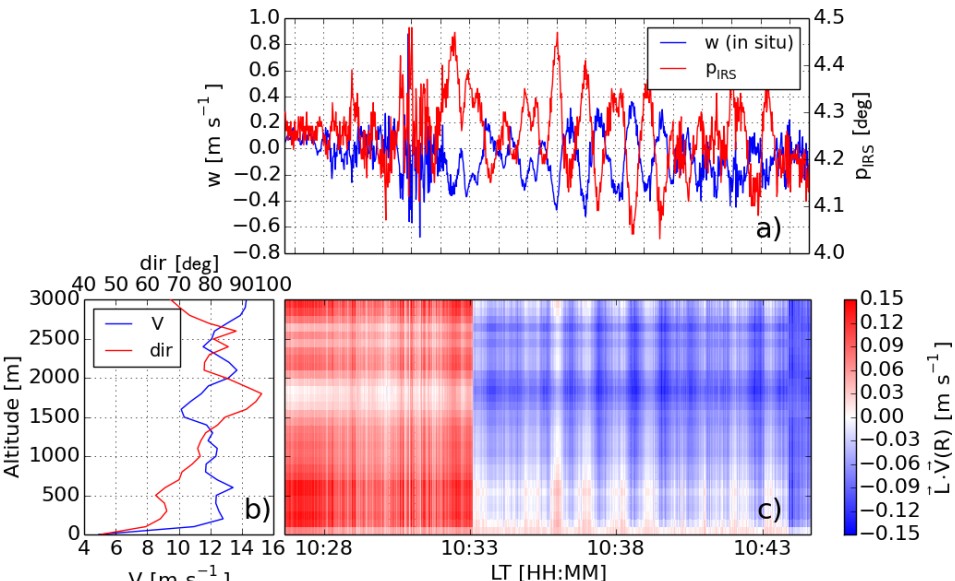

Figure 2. a) Vertical wind speed (blue) measured by the aircraft in-situ sensors at 2900 m,
together with the aircraft pitch (red). b) Average of the horizontal wind speed (blue) and
direction (red) retrieved by the DWL during a previous leg flown along the same track. c)
Projection of the horizontal wind speed to the DWL LOS due to changes in the aircraft
orientation. At 10:33 LT the DWL LOS was changed by the operator in-flight to reduce the
horizontal wind projection.



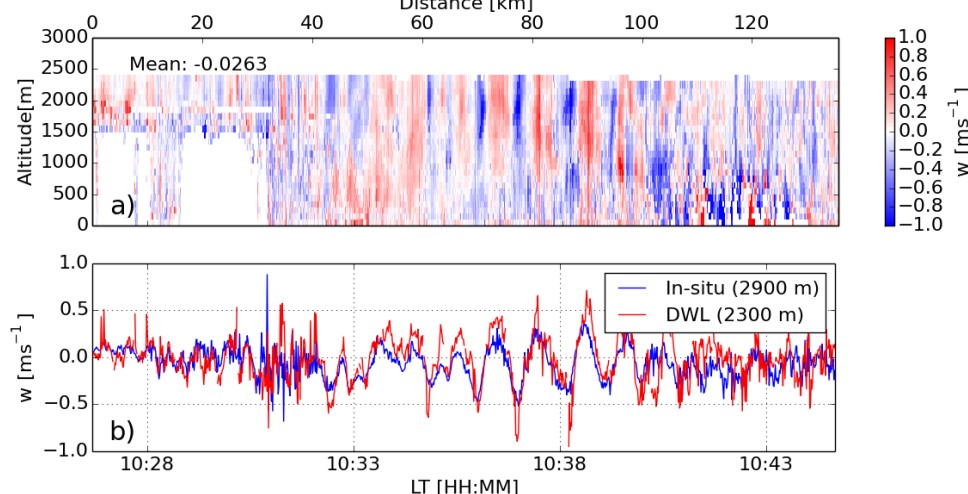

Figure 3. a) DWL vertical wind speed retrieval for a leg flown on 20 June 13 between 10:31
and 10:45 LT. Positive (red) indicates upward winds and negative (blue) indicates downward
winds. Between 10:27 LT and 10:31 LT the DWL coverage is limited due to the presence of
clouds (white regions below 1500 m). b) Comparison between the vertical wind retrieved by
the Falcon 20 (blue) for an altitude of 2900 m and the DWL retrieval (red) for an altitude of
2300 m.



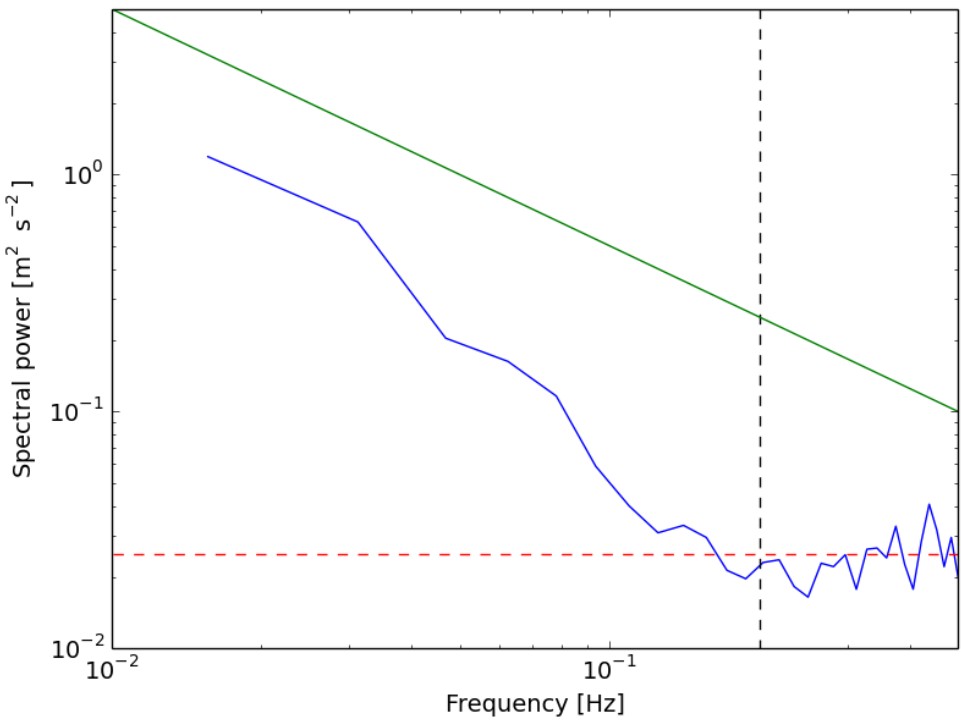

Figure 4. Spectral power for the DWL measured vertical wind speed during the flight on 20
June 2013, between 10:31 and 10:45 LT and for an altitude of 2300 m (solid, blue). The
expected spectral behavior according to the Kolmogorov's -5/3 law (green line), the noise
frequency threshold (black, dotted) and the derived noise floor for the DWL (red, dotted) are
shown.



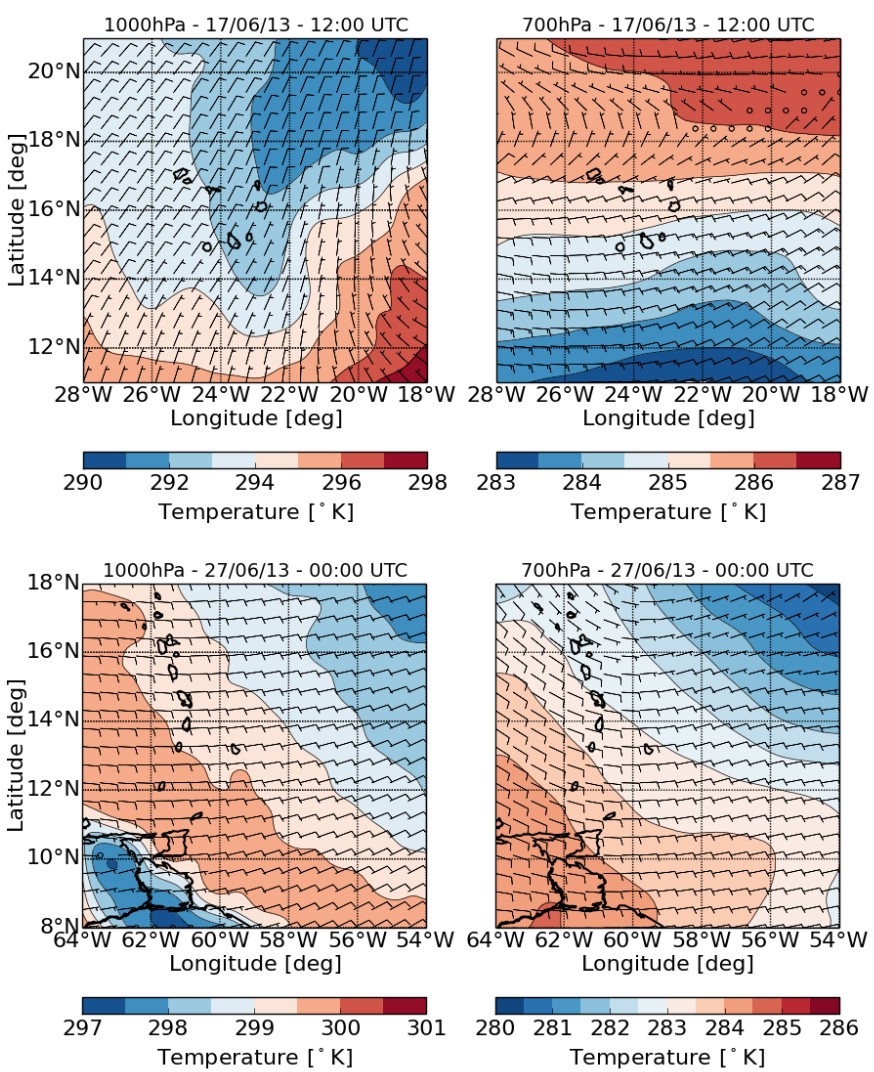

2     Figure 5. Synoptic conditions derived from the ECMWF ERA-Interim reanalysis over the

3     Cape Verde islands (upper panels) and Barbados (lower panels) for the 17 June 2013 and 27

4     June 2013 respectively. Wind vectors (barbs, in m s$^{-1}$) and temperature (in °K) are shown.





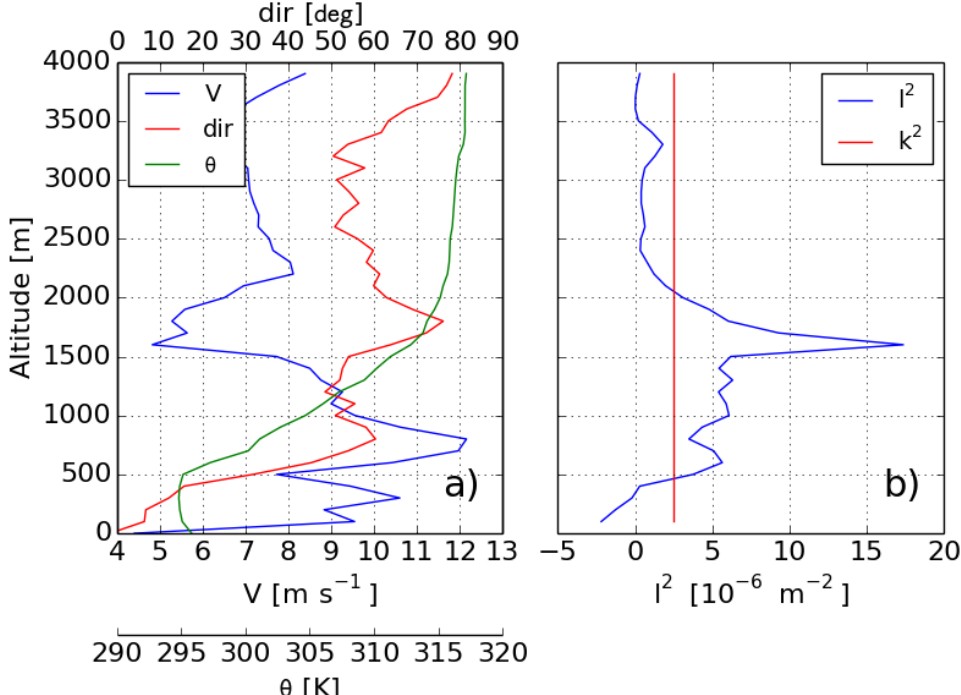

Figure 6. a) Horizontal wind speed (blue), wind direction (red) and potential temperature
(green) measured by the Falcon in-situ sensors during take-off (10:06 to 10:14 LT). b)
Derived Scorer parameter (blue) and approximate wave number corresponding to the
observed waves (red).



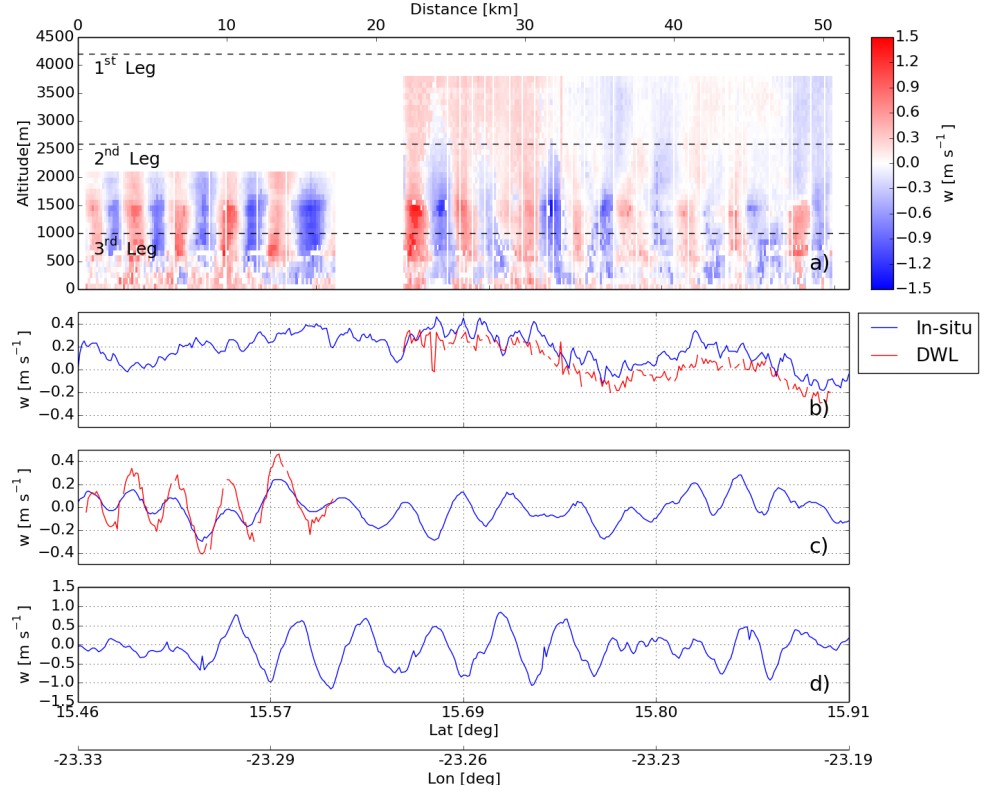

Figure 7. a) Retrieved vertical wind speed by the DWL as a function of the latitude and longitude from legs 1 (10:21 to 10:25 LT) and 2 (10:43 to 10:46 LT), together with the flight levels corresponding to the legs 1, 2 and 3 (11:10 to 11:16 LT) (dashed lines). b, c, d) In-situ vertical wind speed corresponding to the legs 1, 2 and 3 (blue line) together with the measured wind corresponding to the uppermost range gate measured on each leg (red line).





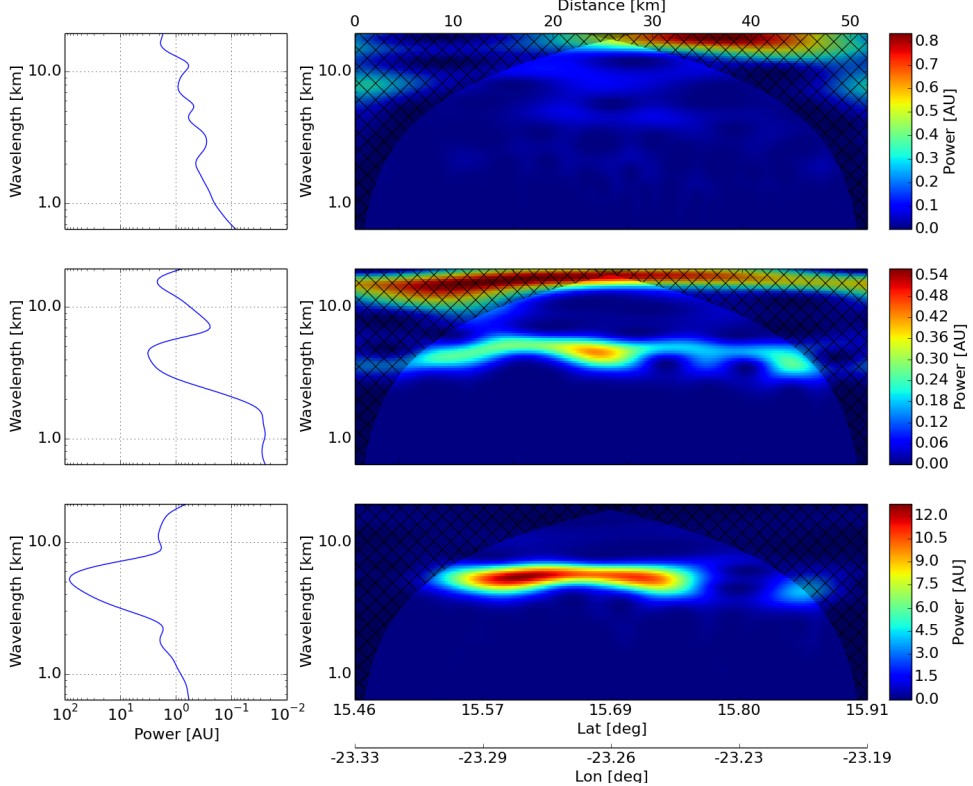

2    Figure 8. In-situ vertical wind wavelet analysis corresponding to the three legs flown on 17

3    June: 1st leg (upper panels), 2nd leg (mid panels) and 3rd leg (lower panels). The hatched areas

4    indicate the cone of influence. The left panels show the average power for each wavelength.



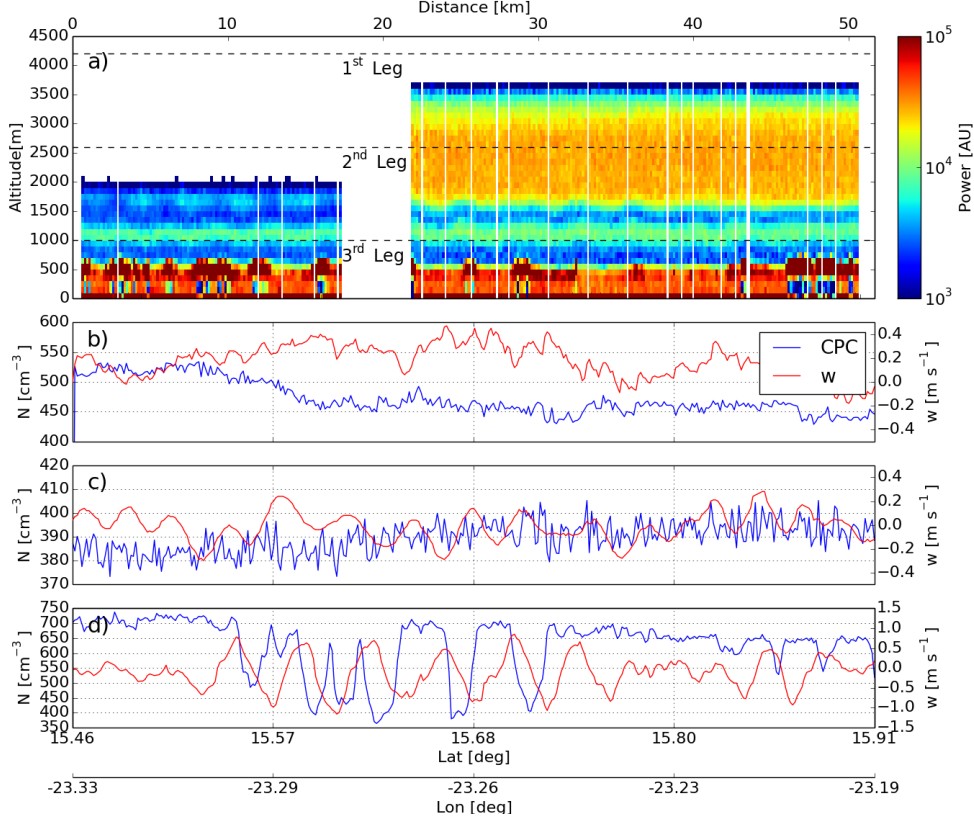

Figure 9. a) Retrieved range corrected backscatter power by the DWL as a function of the latitude and longitude from legs 1 and 2, together with the flight levels corresponding to the legs 1, 2 and 3 (dashed lines). b, c, d) In-situ CPC measurements (blue line) corresponding to the legs 1, 2 and 3 together with the in-situ measured vertical wind (red line).





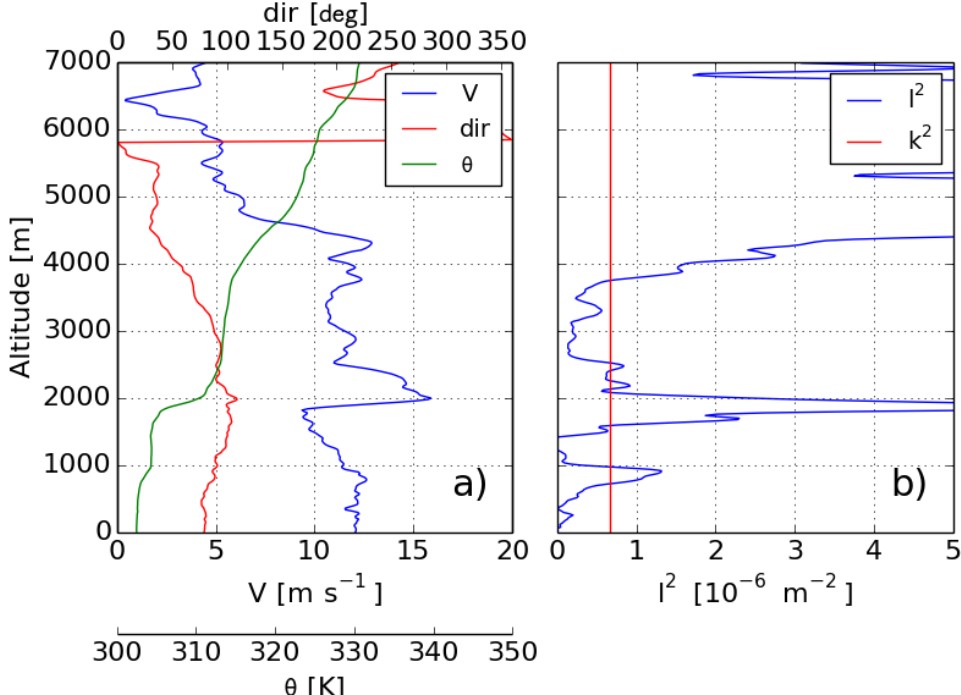

2    Figure 10. a) Horizontal wind speed (blue), wind direction (red) and potential temperature

3    (green) measured by a dropsonde launched at (26.6.13 - 22:05 LT). b) Derived Scorer

4    parameter (blue) and approximate wave number corresponding to the observed waves (red).

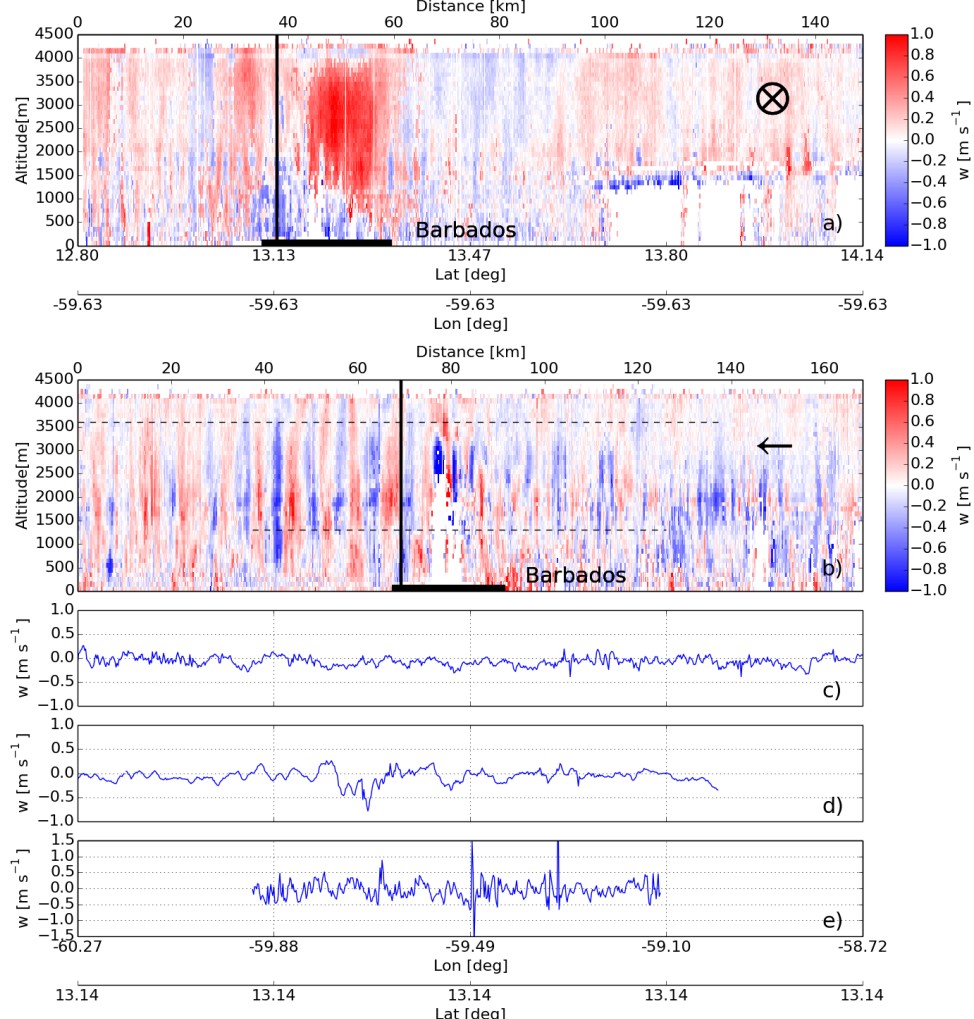

Figure 11. a) Retrieved vertical wind speed by the DWL as a function of the latitude and

longitude from leg 1 (19:53 to 20:06 LT), together with the latitude at which legs 2 (21:49 to

22:05 LT), 3 (22:16 to 22:34 LT) and 4 were flown (22:46 to 23:01 LT) (solid line). b)

Retrieved vertical wind speed by the DWL as a function of the latitude and longitude from leg

2, the intersection position with the $1^{st}$ leg (solid line) and the legs 3 and 4 are shown (dashed

line). c, d, e) In-situ vertical wind speed corresponding to the legs 2, 3 and 4. The average

wind inflow direction is indicated on the upper-right corner of panels a) and b). Barbados

indicated in both plots with a horizontal solid black line.





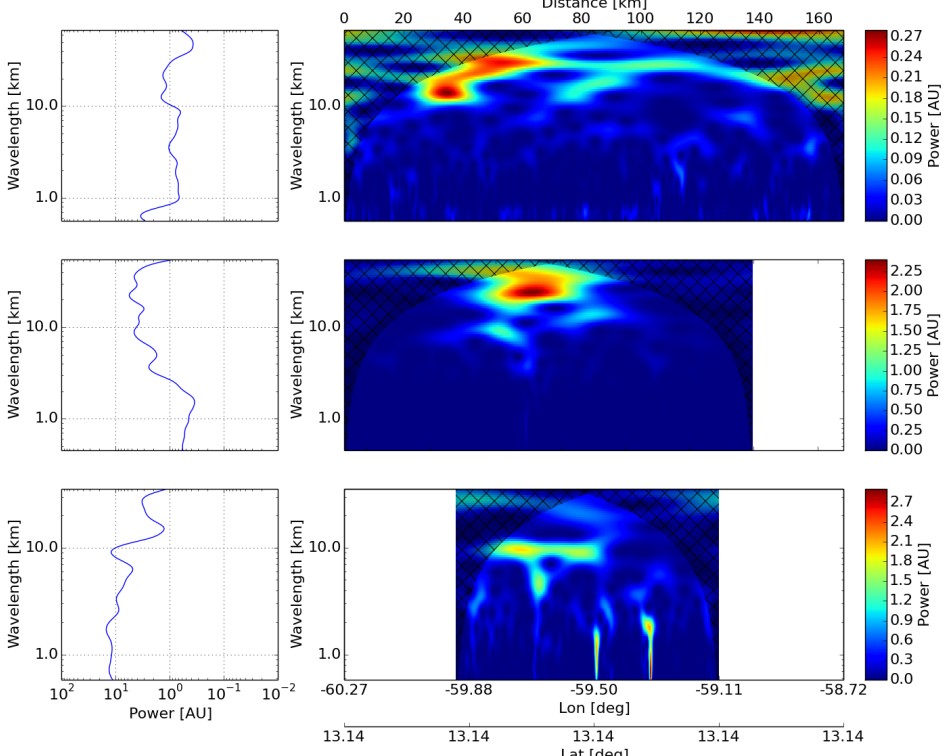

2 Figure 12. In-situ vertical wind wavelet analysis corresponding to three legs flown on 26

3 June: 2<sup>nd</sup> leg (upper panels), 3<sup>rd</sup> leg (mid panels) and 4<sup>th</sup> leg (lower panels). The hatched areas

4 indicate the cone of influence. The left panels show the average power for each wavelength.



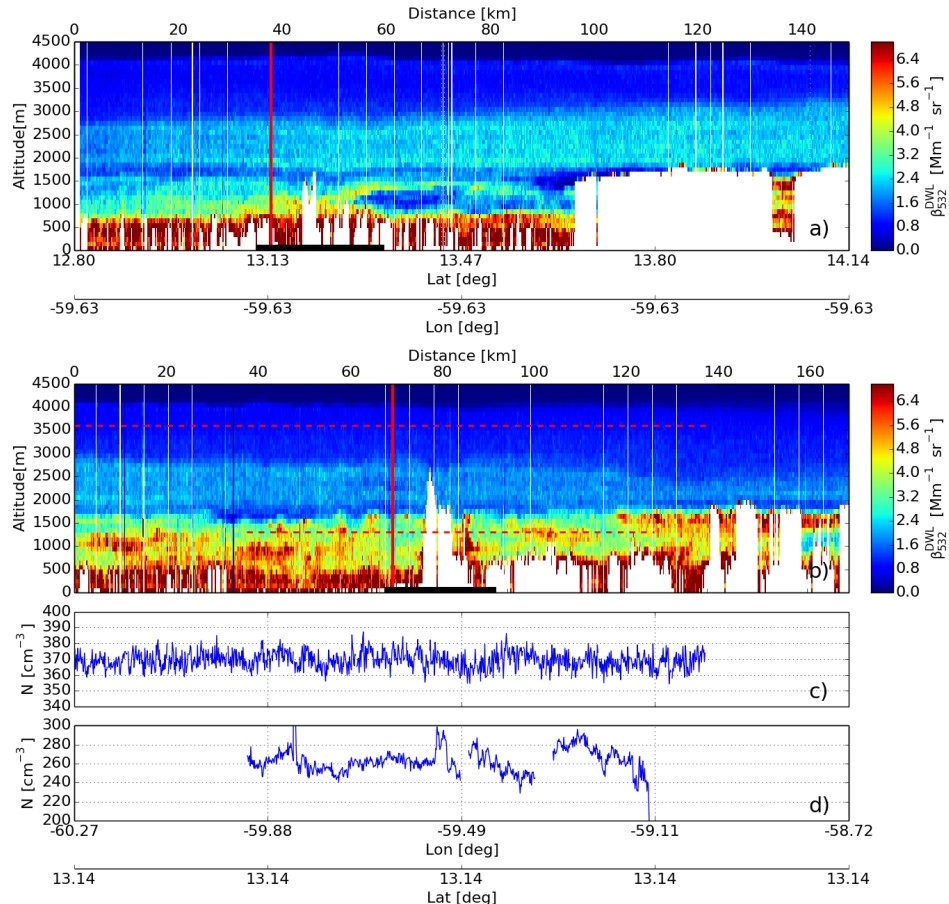

Figure 13. a) Retrieved backscatter coefficient by the DWL as a function of the latitude and
longitude from leg 1, together with the latitude at which legs 2, 3 and 4 were flown (solid line,
red). b) Retrieved backscatter coefficient by the DWL as a function of the latitude and
longitude from leg 2, the intersection position with the 1$^{st}$ leg (solid line, red) and the legs 3
and 4 are shown (dashed line., red). c, d) In-situ CPC measurements corresponding to the legs
3 and 4. The white color indicates regions were no atmospheric signal is available (e.g., below
clouds, unseeded laser operation). Barbados indicated in both plots with a horizontal solid
black line.





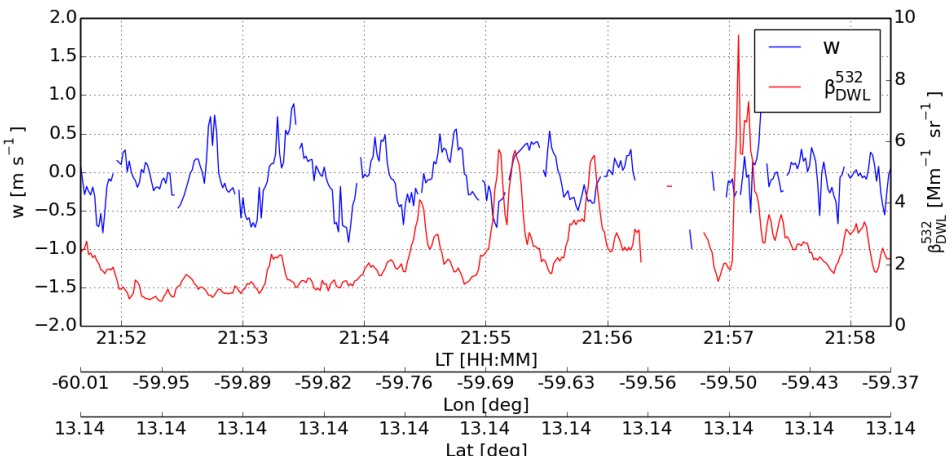

2  Figure 14. Retrieved backscatter coefficient (red) and vertical wind speed (blue) on the lee

3  side of Barbados during the flight of the 2$^{nd}$ leg at altitude of 1600 m. Due to the presence of

4  clouds, data between 21:56 and 21:57 LT is missing.





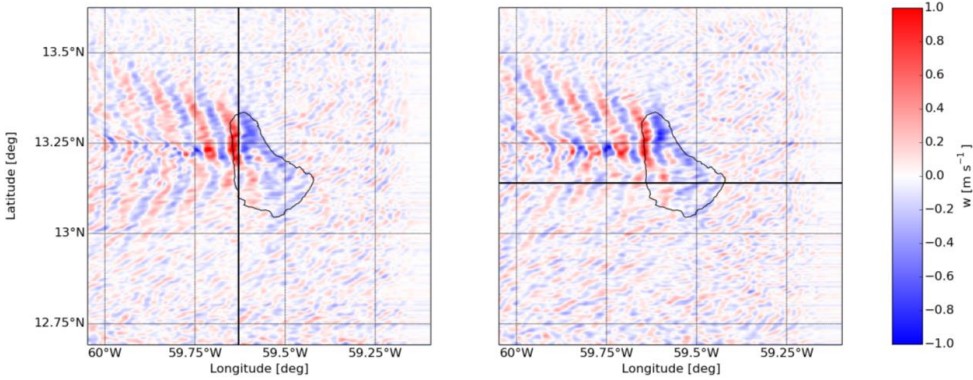

2   Figure 15. Vertical wind speed at 1975 m derived from the LES corresponding to 26 June at

3   20:00 LT (left panel) and 22:00 LT (right panel). The flight tracks corresponding to leg 1

4   (left) and 2 (right) are also indicated (solid, black).





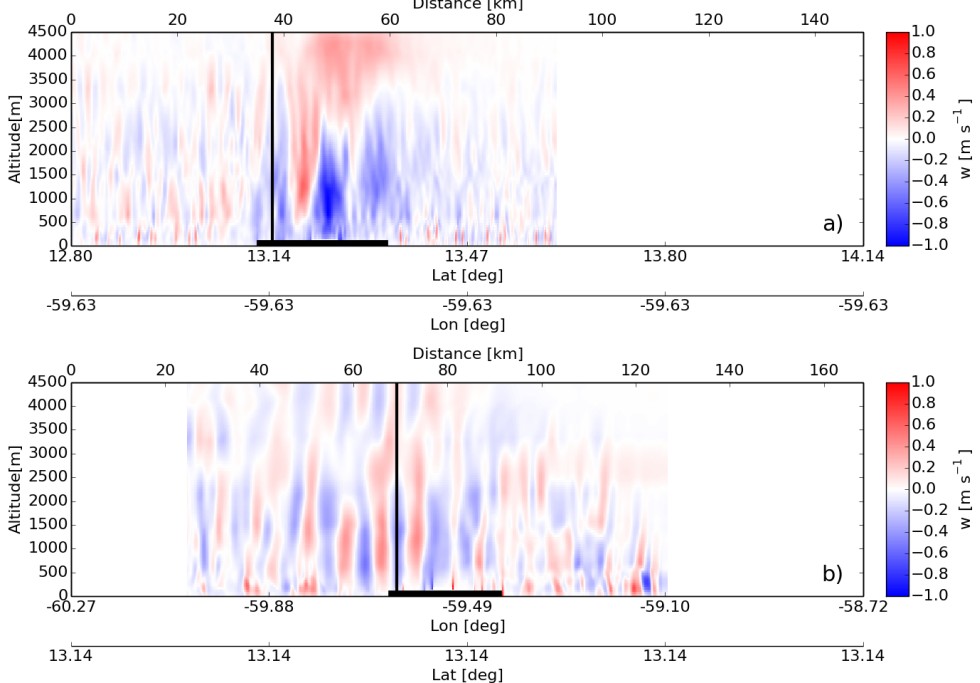

2  Figure 16. Vertical wind speed from the LES for 26 June at 20:00 LT for leg 1 (upper panel)

3  and 22:00 LT for leg 2 (lower panel). The location of Barbados is indicated as a horizontal

4  black segment, while the intersection between legs 1 and 2 is indicated with a vertical line.





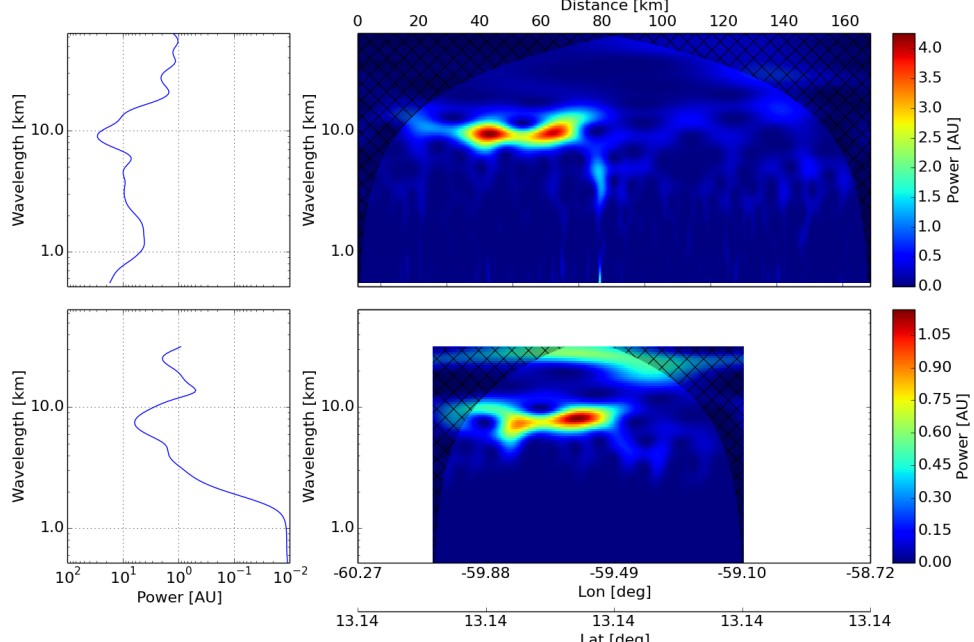

Figure 17. Wavelet analysis of the vertical wind corresponding to the 2$^{nd}$ leg for the flight on

26 June derived from the DWL measurements (Fig. 11, b) at 2000 m (upper panels) and

derived from the LES (Fig. 16, b) at 2000 m (lower panels). The hatched areas indicate the

cone of influence. The left panels show the average power for each wavelength.

