# Peer review of "Vertical wind retrieved by airborne lidar and analysis of"

_Atmospheric Chemistry and Physics, 2015_

## Referee Comment (RC1) · Anonymous Referee #1 · 15 Feb 2016

This paper describes unique observations of vertical wind speed and island induced trapped waves using airborne Doppler lidar during SALTRACE. The paper contains novel and interesting data so that it should be publishable after consideration of the following points: 1. On P5, L9-10, the statement "for the retrieval of vertical 10 wind speed, the LOS vector L âČŮ DWL is set pointing approximately in nadir direction", how is the effect from the airplane movement if the laser beam is not exactly perpendicular to the flight direction? 2. On P11, L15, "Fig. 4" should be Fig. 3b. 3. On P12, L22-24, the statement ". . . lower pressure level. . .For the upper pressure level the wind changes to easterly direction" is confusing according to Fig. 5. The lower pressure level should

be 1000 hPa and the upper pressure level should be 700 hPa. 4. On P15, L21, "cone" should be core. 5. On P16, L15, The phase difference of 90 degrees between the aerosol concentration change and the vertical wind velocity need further explanation. 6. On P18, L28-29, the statement "A wave structure can be identified in the boundary between the SAL and the mixed layer at an altitude of 2000 m." It is not obvious, to my opinion, to recognize a wave pattern, especially in Fig 13a. The leg 2 measurement in Fig 13b does show some wave structure. 7. Table 2., Some estimated mounting angle has large deviation value in rolling and yaw angle, eg. $0.11°$ on 18.06.13, -0.04 on 13.07.13, $0.20°$ and $11.52°$ on 14.07.13. Does it have any influence on surface distance measurement?

---

## Author Comment (AC1) · 24 Feb 2016

Dear reviewer,

We like to thank you for your helpful comments on our paper titled "Vertical wind retrieved by airborne lidar and analysis of island induced gravity waves in combination with numerical models and in-situ particle measurements".

The original comments are in bold, followed by our replies.

**Reviewer #1**

1. **On P5, L9-10, the statement "for the retrieval of vertical wind speed, the LOS vector L_DWL is set pointing approximately in nadir direction", how is the effect from the airplane movement if the laser beam is not exactly perpendicular to the flight direction?**

   A lidar pointing direction $\vec{L}$ not perpendicular to the flight direction introduces a projection of the horizontal aircraft speed in the DWL measurement equal to $\vec{L} \cdot \vec{v}_{ac}$ and a projection of the horizontal wind component equal to $\vec{L} \cdot \vec{V}(\mathbf{R})$. For the case of the aircraft speed projection, a correction based on the aircraft navigation system measurements and the mounting angle of the lidar is applied according to Eq. 2, while for the case of the atmospheric horizontal wind speed, previous DWL horizontal wind speed measurements and dropsonde wind speed profiles can be used to partially compensate this effect (Fig. 2).

   The following clarification was introduced:

   *"Corrections to the LOS speed resulting from a non-zero nadir angle are discussed in Sec. 2.3."*

2. **On P11, L15, "Fig. 4" should be Fig. 3b.**

   Fig. 4 was changed to Fig. 3b.

3. **On P12, L22-24, the statement "...lower pressure level. For the upper pressure level the wind changes to easterly direction" is confusing according to Fig. 5. The lower pressure level should be 1000 hPa and the upper pressure level should be 700 hPa.**

Clarification introduced:

*According to the reviewer suggestion, "lower" was changed by "1000 hPa" and "upper" by "700 hPa".*

**4. On P15, L21, "cone" should be core.**

The terminology related to the wavelet analysis follows the work by Torrence and Compo: "A practical guide to wavelet analysis", included as a reference in this work.

**5. On P16, L15, The phase difference of 90 degrees between the aerosol concentration change and the vertical wind velocity need further explanation.**

The following clarifications were introduced:

*"…the CPC measurements show also the effect of the waves in the aerosol distribution, but with a phase difference to the vertical wind speed of 90°"*

*"Because the aerosol vertical flux is determined by the integrated product of the vertical wind speed and the variation of the aerosol concentration with respect to the mean along the flight path, a phase difference of 90 degrees between these two quantities results in a zero net flux. The dust loaded air parcels periodically rise and sink without leading to a net downward or upward dust transport."*

**6. On P18, L28-29, the statement "A wave structure can be identified in the boundary between the SAL and the mixed layer at an altitude of 2000 m." It is not obvious, to my opinion, to recognize a wave pattern, especially in Fig 13a. The leg 2 measurement in Fig 13b does show some wave structure.**

The sentence refers to the wave pattern visible in Fig. 13b.

Clarification introduced:

*"On the 2nd leg, a wave structure can be identified in the boundary between the SAL and the mixed layer at an altitude of 2000 m on the lee side of Barbados (Fig. 13b)."*

**7. Table 2., Some estimated mounting angle has large deviation value in rolling and yaw angle, eg. 0.11° on 18.06.13, -0.04 on 13.07.13, 0.20° and 11.52° on 14.07.13. Does it have any influence on surface distance measurement?**

As can be seen in the geometry of the problem presented in Fig. 1 and in P9 L11-18, deviations in the yaw mounting angle does not have a large influence in the surface distance measurement. For that reason, the yaw mounting angle estimation based on distance measurements (second row of each flight presented in Table 2) give place to an unstable solution. A similar behavior is observed for the roll angle in the case of mounting angle derivations based on speed measurement. A combination of both methods gives place to more stable solutions.

---

## Referee Comment (RC2) · Anonymous Referee #2 · 23 Mar 2016

Review on ACP

"Vertical wind retrieved by airborne lidar and analysis of island induced gravity waves in combination with numerical models and in-situ particle measurements" by F. Chouza et al., submitted to ACPD.

General:

Excellent paper introducing the advantages of airborne LIDAR measurements investigating gravity waves on the basis of measurements of the vertical wind velocity and backscatter coefficient, validation by in-situ measurements and comparison to LES modeling.

The experiment is described well. Two case studies are well analyzed and validated.

Tools: LES modeling, wavelet analysis, comparison to in-situ measurements of meteorological parameters and aerosols.

Interesting results relying on the topic of the special issue of ACP and ACP at all.

For the scientific community new interesting case studies on airborne LIDAR application.

Review recommendation points 1-12 are fulfilled.

Number and quality of references is good.

Minor technical comments

> Page 2 line 23-27 sentence too long split off into two or three sentence to clarify the statement
> Page 5 line 13-17 split off the sentence
> Page 5 error introduced by the horizontal wind into the vertical wind component
> Page 6 time resolution of angle measurements
> Page 9: Eq (7) is missing
> Page 9: line 3-6 split off the sentence
> Page 11 line 15: fig. 3b instead off fig. 4
> Page 12:  line 26 fig S1=?
> Page 13:  please introduce first the Scorer-parameter, (Eq. 7) earlier (at line 5)
> Page 16: line 23:  fig S2=?
>  Figure S1 and S2 are not clear
> Page 19: line 9: what is meant by low humidity conditions? (free troposphere?) please specify

> Figure S1 and S2 are not clear
> Figure 1:  abbreviation DEM?

---

## Author Comment (AC2) · 29 Mar 2016

Dear reviewer,

We like to thank you for your helpful comments on our paper titled "Vertical wind retrieved by airborne lidar and analysis of island induced gravity waves in combination with numerical models and in-situ particle measurements".

The original comments are in bold, followed by our replies and a marked-up version of the manuscript indicating the changes introduced according to the answers .

**Reviewer #2**

1. **Page 2 line 23-27 sentence too long split off into two or three sentence to clarify the statement**

   The sentence was reformulated in the following way:

   *"During the SAMUM-2 campaign, a large eddy simulation (LES) study was performed in the Cape Verde region (Engelmann et al., 2011). This study showed that a flat island with the characteristics of the Santiago Island (Cape Verde) can induce the generation of gravity waves and enhance the aerosol downward mixing, only through its heat island effect, without taking into account its orography."*

2. **Page 5 line 13-17 split off the sentence**

   The sentence was reformulated in the following way:

   *"Because the DWLs measure relative speed between the instrument and the sensed atmospheric volume, the retrieval of the wind speed requires knowing the aircraft speed component of the measurement. This calculation requires especially accurate measurements of the aircraft speed, orientation, and DWL relative position with respect to the aircraft."*

3. **Page 5 error introduced by the horizontal wind into the vertical wind component**

   The error introduced by the horizontal wind is discussed in the section 2.3. The following clarification was introduced:

*"Corrections to the LOS speed resulting from a non-zero nadir angle are discussed in Sec. 2.3."*

**4. Page 6 time resolution of angle measurements**

The following clarification was introduced:

*"The Falcon IRS system provides yaw angle measurements at a rate of 20 Hz, and pitch and roll measurements at 50 Hz. These measurements are then averaged for 1 s to match the DWL accumulation time."*

**5. Page 9: Eq (7) is missing**

The equation number was wrong. Eq. (7) was changed to Eq. (6) and Eq. (6) to Eq. (5).

**6. Page 9: line 3-6 split off the sentence**

The sentence was rearranged as follows:

*"For each flight, surface returns (land and sea) corresponding to measurements in scanning operation mode were used to retrieve the mounting angle. Only the retrievals corresponding to flight altitudes higher than 5000 m and with vertical aircraft speeds lower than 0.05 m s-1 where included in the calculation."*

**7. Page 11 line 15: fig. 3b instead off fig. 4**

Fig. 4 was changed to Fig. 3b.

**8. Page 12: line 26 fig S1=?**

Fig. S1 refers to the first supplemental figure.

**9. Page 13: please introduce first the Scorer-parameter, (Eq. 7) earlier (at line 5)**

Fig. 6b (Scorer parameter plot) is now introduced after the theoretical introduction of the Scorer parameter.

**10. Page 16: line 23: fig S2=?**

Fig. S2 refers to the second supplemental figure.

**11. Page 19: line 9: what is meant by low humidity conditions? (free troposphere?) please specify**

The following clarification was introduced:

*"Dry atmospheric conditions are required in order to avoid the hygroscopic growth of aerosol particles, which otherwise would lead to a wrong flux estimation. The exact relative humidity threshold under which hygroscopic growth can be neglected varies according to the aerosol type. For the case of sea salt and desert dust, hygroscopic growth can be safely neglected for a relative humidity below 50% (e.g. Engelmann et al., 2008; Kaaden et al., 2008)."*

**12. Figure S1 and S2 are not clear**

After trying different ways to present the flight paths and measurements, we decided to leave it as it is. Although a 3D view of the flight track adds complexity, a 2D plot cannot capture the different flight levels and changes in the measurement pattern.

**13. Figure 1: abbreviation DEM?**

The following clarification was introduced in the Fig. 1 caption:

*"DEM is Digital Elevation Model."*

**References**

*Engelmann, R., Wandinger, U., Ansmann, A., Müller, D., Žeromskis, E., Althausen, D. and Wehner, B.: Lidar Observations of the Vertical Aerosol Flux in the Planetary Boundary Layer. J. Atmos. Oceanic Technol., 25, 1296–1306, doi: http://dx.doi.org/10.1175/2007JTECHA967.1, 2008.*

*Kaaden, N., Massling, A., Schladitz, A., M¨uller, T., Kandler, K., Schütz, L., Weinzierl, B., Petzold, A., Tesche, M., Leinert, S., Deutscher, C., Ebert, E., Weinbruch, S., and Wiedensohler, A.: State of mixing, shape factor, number size distribution, and hygroscopic growth of the Saharan anthropogenic and mineral dust aerosol at Tinfou, Morocco, Tellus, 1600–0889, 2008.*